# When to Trust the Cheap Check: Weak and Strong Verification for Reasoning

**Shayan Kiyani** [1]  **Sima Noorani** [1]  **George Pappas** [1]  **Hamed Hassani** [1]

## Abstract

Reasoning with LLMs increasingly unfolds inside a broader verification loop. Internally, systems use cheap checks, such as self-consistency or proxy rewards, which we call *weak verification*. Externally, users inspect outputs and steer the model through feedback until results are trustworthy, which we call *strong verification*. These signals differ sharply in cost and reliability: strong verification can establish trust but is resource-intensive, while weak verification is fast and scalable but noisy and imperfect. We formalize this tension through *weak–strong verification policies*, which decide when to accept or reject based on weak verification and when to defer to strong verification. We introduce metrics capturing incorrect acceptance, incorrect rejection, and strong-verification frequency. Over population, we show that optimal policies admit a two-threshold structure and that *calibration* and *sharpness* govern the value of weak verifiers. Building on this, we develop an online algorithm that provably controls acceptance and rejection errors without assumptions on the query stream, the language model, or the weak verifier. Experiments on mathematical reasoning and sequential decision-making demonstrate that our algorithm achieves reliability comparable to exhaustive strong verification while significantly reducing verification cost.

## 1. Introduction

Recent advances in the reasoning capabilities of large language models have been driven not only by larger models or increased test-time computation, but critically by the incorporation of verification into the inference process. In practice, verification arises on two complementary fronts:

On the user side, model outputs are treated as high-potential candidates that are subject to careful evaluation. This evaluation may involve inspecting outputs line by line or, depending on the domain, testing them externally in the real world. Crucially, this process is informed by context, preferences, and domain knowledge that may extend beyond what can be captured textually. While this form of verification enforces the highest level of trust, it is inherently costly; we refer to it as *strong verification*.

On the model side, verification is engineered to operate at scale, with the goal of approximating the judgments users make under strong verification. This includes self-consistency, learned critiques, or proxy rewards that attempt to anticipate what users would agree with. Additionally, in some domains, specialized tools can provide fast, local checks of correctness (e.g., by executing code), but these typically certify narrow aspects of an output rather than whether a line of reasoning is promising overall, as an expert would judge. We refer to this form of fast, scalable verification as *weak verification*.

This contrast highlights a fundamental tension: the mechanisms that most effectively establish trust are costly to deploy at scale, while the mechanisms that scale effortlessly are often insufficient to support reliable use on their own. Therefore, what is missing in current reasoning systems is a principled way to orchestrate verification sources. We ask:

*Can we match the reliability we would get if strong verification were applied at every step, while deploying it on only a small, carefully chosen fraction of the reasoning process?*

We answer this question affirmatively by developing a principled framework in which internal verification signals are used not only to refine candidate solutions, but also to inform when stronger verification should be called. Our core contribution is a novel online calibration algorithm, Selective Strong Verification (SSV), that explicitly controls the mismatch between weak and strong verification signals on the subset of instances where decisions are made solely based on weak verification, without consulting the strong signal. This enables strong verification resources to be deployed selectively, while preserving reliable end-to-end behavior. We now outline our contributions and the structure of the paper.

[1]Department of Electrical and Systems Engineering, University of Pennsylvania, USA. Correspondence to: Shayan Kiyani <shayank@seas.upenn.edu>.

*Proceedings of the 43rd International Conference on Machine Learning*, Seoul, South Korea. PMLR 306, 2026. Copyright 2026 by the author(s).

(1) In Section 3, we formalize *weak–strong verification policies*: an algorithmic framework that governs when a system should rely solely on a weak verifier and when it should defer to a strong verifier. We introduce three performance metrics that capture the fundamental tradeoffs in this setting. The first two are *type-I* and *type-II* errors, capturing incorrect trust in the weak verifier when it disagrees with the strong verifier: accepting incorrect responses or rejecting correct ones. The third metric is the frequency with which the strong verifier is queried.

(2) In Section 4, we characterize optimal weak–strong verification policies under population assumptions and show that they admit a two-threshold structure, which underpins our algorithm. Along the way, our analysis highlights *calibration* and *sharpness* as two key properties governing the effectiveness of weak verifiers, a perspective that may be of independent interest.

(3) In Section 5, we develop SSV, a finite-sample, online algorithm that provably controls type-I and type-II errors. Our guarantees are fully distribution-free and make no assumptions on the query stream, the behavior of the language model, or the quality of the weak verifier.

(4) In Section 6, we evaluate SSV on outcome-level mathematical reasoning and process-level sequential puzzle solving. We demonstrate that our algorithm maintains target error rates in finite-samples while enabling principled navigation of the reasoning-verification cost frontier, achieving near-oracle accuracy with significantly fewer costly calls.

## 2. Related Works

**LLM reasoning and verification.** Recent progress in LLM reasoning is driven largely on two main axes. First, a large body of work focuses on improving the reasoning process via inference-time reasoning, including structured prompting (Wei et al., 2023; Yao et al., 2023a;b), search (Xie et al., 2024; 2023), decoding strategies (Sun et al., 2024; Xia et al., 2023), as well as training approaches that elicit longer reasoning chains (Wang et al., 2025; Shao et al., 2024). These methods treat weak signals as fixed and strategize reasoning traces to improve final performance.

Second, complementary work improves the weak verification signal itself, including LLM-as-judge evaluation (Liu et al., 2023), specialized verifiers (Saad-Falcon et al., 2024; Tang et al., 2024), judge-time scaling (Kalra & Tang, 2025; Saad-Falcon et al., 2025), and process-reward models (Wang et al., 2024; Zhang et al., 2025; Wang et al., 2023). Our work is orthogonal to both: we take the weak verifier and reasoning procedure as fixed, and study the layer above, orchestrating when to trust the weak signal and when to invoke costly strong verification. This framework applies to any reasoning procedure (single pass, iterative refine-ment, or tree search) and any scoring model. To the best of our knowledge, this interaction has not been explicitly formulated or analyzed.

**Selective prediction and learning to defer.** Algorithmically, our setup relates to selective prediction and learning-to-defer (L2D). Early work established theoretical frameworks for classification with a reject option, posing the problem as risk minimization with explicit rejection costs (Bartlett & Wegkamp, 2008; El-Yaniv & Wiener, 2010). Rather than fixing confidence thresholds post hoc, subsequent work learns when to abstain as part of training (Cortes et al., 2023; Geifman & El-Yaniv, 2017; Gangrade et al., 2021), with extensions to the online setting (Gangrade et al., 2021). The L2D literature extends selective prediction to human-AI collaboration, studying the optimal division of labor between model and expert (Mozannar & Sontag, 2021; Okati et al., 2021; Verma & Nalisnick, 2022; Mozannar et al., 2023). Our setting can be viewed as an instance of L2D, where deferral means invoking strong verification. The combination of distribution-free online calibration, partial feedback, and separate Type-I/II error control, together with the algorithmic techniques we develop, may be of independent interest to the broader umbrella of L2D.

## 3. Weak–Strong Verification Policies and Metrics

We consider a general verification-guided reasoning setting involving a language model and two sources of verification.

**Language model and verification oracles.** Let $P \in \mathcal{P}$ denote a problem instance or prompt that requires reasoning. Let $f : \mathcal{P} \to \mathcal{R}$ denote a language model that, given $P$, generates a (possibly random) response $R := f(P)$.

We consider two forms of verification. First, let $g : \mathcal{P} \times \mathcal{R} \to \{0, 1\}$ denote a *strong verification* oracle, which outputs a binary judgment indicating whether a response is correct. This oracle represents the strongest form of verification available, such as human inspection or domain specific executions, and serves as the ultimate criterion against which reasoning outcomes are evaluated.

Second, let $w : \mathcal{P} \times \mathcal{R} \to [0, 1]$ denote a *weak verification oracle*, which assigns a real-valued score to a prompt–response pair, aiming to approximate strong verification, for example through proxy rewards or domain-specific tools. The continuous nature of $w$ reflects uncertainty: it provides a confidence signal, with larger values indicating greater confidence in correctness.

**Stream of queries and responses.** We assume there exists an arbitrary and unknown stream of queries. At each time step $t = 1, 2, \ldots$, the language model receives a query $P_t$

and produces a response $R_t = f(P_t)$. We use the notation $w_t := w(P_t, R_t)$ and $g_t := g(P_t, R_t)$.

We place no assumptions on how the stream $\{P_t\}_{t \geq 1}$ is generated. In particular, queries may be independent user prompts, intermediate reasoning steps, or any combination thereof, and the stream may depend arbitrarily on past verification outcomes. This modeling is flexible enough to capture a range of reasoning strategies. For example,

- Each $P_t$ may correspond to a user prompt, each $R_t$ to a full model output. This resembles to a strategy known as output reward modeling in the literature (Cobbe et al., 2021).

- In step-by-step reasoning, each $P_t$ may consist of a prompt together with the partial solution so far, and each $R_t$ corresponds to a single reasoning step. This resembles process reward modeling in the literature (Lightman et al., 2023).

In both cases, rejecting/accepting responses can influence future queries, either by triggering another sample for the same prompt until a budget is exhausted, or by moving on to a different prompt.

**Weak-strong verification policy.** A weak–strong verification policy is a sequence of (possibly randomized) functions

$$\pi_t(\cdot) : [0, 1] \rightarrow \{A, R, SV\}, \quad \forall\, t \geq 1,$$

which maps a weak verification score to one of three actions:

**A:** accept the response without asking for strong verification.

**R:** reject the response without asking for strong verification.

**SV:** query the strong verifier; accept the response if $g(P, R) = 1$ and reject it otherwise.

At iteration $t = 1, 2, \ldots$, the system proceeds as follows:

- The language model generates a response $R_t = f(P_t)$.

- The weak verifier is queried, yielding a score $w_t$.

- The policy $\pi_t(w_t)$ determines whether to accept or reject the response, with or without calling strong verification.

The policy $\{\pi_t\}_{t>0}$ thus governs when the weak signal is sufficient to accept or reject a response, and when the algorithm defers to strong verification $g_t$. Figure 1 depicts the overall flow.

**Performance metrics.** We evaluate a verification policy along the interaction sequence, without imposing any distributional assumptions on the stream $\{(P_t, R_t)\}_{t \geq 1}$. For a fixed horizon $T \geq 1$, define the index sets

$$\mathcal{S}_0(T) := \{t \leq T : g_t = 0\}, \quad \mathcal{S}_1(T) := \{t \leq T : g_t = 1\},$$

and their sizes $N_0(T) := |\mathcal{S}_0(T)|$ and $N_1(T) := |\mathcal{S}_1(T)|$. We emphasize that $g_t$ always exists conceptually, but is only observed when the strong verifier is queried.

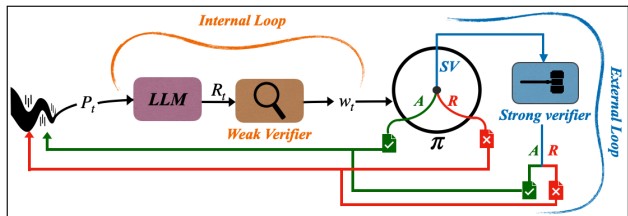

*Figure 1.* The architecture of weak-strong verification for LLM reasoning.

We define the following empirical performance quantities:

**Type-I error (incorrect acceptance):**[1]

$$\text{Err}^{\text{I}}(T) := \frac{1}{N_0(T)} \sum_{t=1}^{T} \mathbf{1}\{g_t = 0,\ \pi_t(r_t) = A\}.$$

This is the empirical frequency of accepting a response among those rounds where the strong verifier would deem it incorrect, i.e., acceptance errors *conditioned on* $g_t = 0$.

**Type-II error (incorrect rejection):**

$$\text{Err}^{\text{II}}(T) := \frac{1}{N_1(T)} \sum_{t=1}^{T} \mathbf{1}\{g_t = 1,\ \pi_t(r_t) = R\}.$$

This is the empirical frequency of rejecting a response among those rounds where the strong verifier would deem it correct, i.e., rejection errors *conditioned on* $g_t = 1$.

**Strong verification frequency:**

$$\text{SV}(T) := \frac{1}{T} \sum_{t=1}^{T} \mathbf{1}\{\pi_t(r_t) = SV\}.$$

This is the average rate at which the algorithm calls strong verification and thus reflects verification cost or latency.

Type-II error and strong verification frequency impose additional system load, such as increased user latency or higher computational and operational costs. In contrast, type-I error corresponds to degraded output quality, manifesting as hallucinations or incorrect responses, which reduces system value and erodes trust among downstream decision makers.

A good verification policy must therefore seek to keep all three quantities small. However, there are inherent tradeoffs between them. For a fixed language model, weak verifier, and strong verifier, enforcing sufficiently small type-I and type-II error rates may necessarily require a higher frequency of strong verification calls. This leads to an engineering tradeoff on the service provider side: given the quality of the language model and the available verification signals, how should one prioritize these competing objectives?

---

[1]If $N_0(T) = 0$ (respectively, $N_1(T) = 0$), we set $\text{Err}^{\text{I}}(T) = 0$ (respectively, $\text{Err}^{\text{II}}(T) = 0$).

In the next section, we first take a different route by studying the Pareto-optimal tradeoff achievable by verification policies over population. This allows us to derive the structure of optimal policies and to motivate specific algorithmic design choices. Building upon that, we then turn to the the goal of controlling type-I and type-II errors while minimizing the use of strong verification in the sequential setting.

## 4. Fundamental Tradeoffs of Weak-Strong Verification

Here we study the fundamental tradeoffs among the three quantities introduced in Section 3 through a simplified, population-level formulation. This is not intended to address the sequential problem of Section 3, but rather to isolate a one-shot version of the problem where tradeoffs can be characterized explicitly. The insights from this population perspective will guide the design of the *sequential, distribution-free algorithm* developed in the next section.

**Population-level formulation.** We consider a one-shot setting in which a prompt–response pair $(P, R)$ is drawn from a joint distribution $\mathcal{D}$, with $R = f(P)$. The language model $f$, weak verifier $w : \mathcal{P} \times \mathcal{R} \to [0, 1]$, and strong verifier $g : \mathcal{P} \times \mathcal{R} \to \{0, 1\}$ are treated as fixed.

A weak–strong verification policy is a mapping $\pi : [0, 1] \to \{A, R, SV\}$, which assigns an action based on the weak score $w(P, R)$. Time indices are unnecessary in this one-shot setting.

**Population metrics.** We define probabilistic counterparts of the empirical quantities from Section 2:

- **Type-I error:** $\Pr_{\mathcal{D}}[\pi(w(P, R)) = A \mid g(P, R) = 0]$.

- **Type-II error:** $\Pr_{\mathcal{D}}[\pi(w(P, R)) = R \mid g(P, R) = 1]$.

- **Strong verification rate:** $\Pr_{\mathcal{D}}[\pi(w(P, R)) = SV]$.

These metrics abstract away temporal dependence and allow us to study intrinsic tradeoffs in expectation.

**Pareto-optimal tradeoffs.** For $\lambda_1, \lambda_2 \geq 0$, we study

$$\pi^\star_{(\lambda_1, \lambda_2)} \in \arg\min_{\pi} \ \Pr[\pi(w(P, R)) = SV] \qquad (1)$$
$$+ \lambda_1 \ \Pr[\pi(w(P, R)) = A \mid g(P, R) = 0]$$
$$+ \lambda_2 \ \Pr[\pi(w(P, R)) = R \mid g(P, R) = 1].$$

Sweeping $(\lambda_1, \lambda_2)$ varies the relative penalty assigned to type-I errors, type-II errors, and strong-verification calls, thereby encoding different operating points (e.g., prioritize reliability vs. latency/cost). The resulting optimizers $\pi^\star_{(\lambda_1, \lambda_2)}$ trace the set of Pareto-optimal policies, revealing which performance levels are achievable.

**Assumption 4.1** (Calibration of the weak verifier). The weak verification oracle $r$ is calibrated with respect to the strong verifier $g$ in the following sense: for all $p \in [0, 1]$,

$$\Pr\big[g(P, R) = 1 \mid w(P, R) = p\big] = p,$$

where $(P, R)$ is distributed as $P \sim \mathcal{D}$ and $R = f(P)$.

Assumption 4.1 is a natural starting point: without some link between the weak score $w(P, R)$ and the correctness signal $g(P, R)$, the weak verifier could in principle be arbitrary and carry no actionable information. While we do *not* rely on calibration in the next section, it provides a clean population model in which we can derive structural insights about how weak and strong verification should interact.

**Theorem 4.2** (Structure of an optimal policy). *Suppose Assumption 4.1 holds. For any $\lambda_1, \lambda_2 \geq 0$, there exists an optimal policy $\pi^\star(\lambda_1, \lambda_2)$ that has a threshold structure: there exist thresholds $t_{\mathrm{low}}, t_{\mathrm{high}} \in [0, 1]$ such that*

$$\pi^\star(w) \in \begin{cases} \{R\}, & w < t_{\mathrm{low}}, \\ \{SV\}, & t_{\mathrm{low}} \leq w \leq t_{\mathrm{high}}, \\ \{A\}, & w > t_{\mathrm{high}}. \end{cases}$$

Theorem 4.2 aligns with the following intuition: when the weak verifier is sufficiently confident that a response is incorrect (small $w$), it is optimal to reject; when it is sufficiently confident that a response is correct (large $w$), it is optimal to accept; and when the weak signal is ambiguous (intermediate $w$), it is optimal to defer to strong verification. In particular, calibration makes the weak score directly interpretable as a probability of correctness, which in turn makes it easy to translate into A/R/SV decisions.

However, calibration is not the only relevant property of a weak verifier. Even a calibrated weak verifier can be unhelpful if it is *uninformative*. For example, a verifier that always outputs the marginal probability $\Pr[g(P, R) = 1]$ is perfectly calibrated, yet carries no instance-specific information and is therefore uninformative.

**What properties of the weak verifier matter?**

The next proposition characterizes the value of the optimal objective and makes precise how the distribution of weak scores governs the achievable tradeoff.

**Proposition 4.3** (Value of the optimal objective). *Assume Assumption 4.1. Let $W := w(P, R) \in [0, 1]$, $\alpha_0 := \Pr[g(P, R) = 0]$ and $\alpha_1 := \Pr[g(P, R) = 1]$. Then the minimum value of the objective in (1) can be written as*

$$V(\lambda_1, \lambda_2) = \mathbb{E}\left[\min\left\{1, \frac{\lambda_1}{\alpha_0}(1 - W), \frac{\lambda_2}{\alpha_1} W\right\}\right].$$

*Interpretation.* The three terms inside the min correspond to the three actions at a given weak score $W$: *(i)* calling strong

verification costs 1; *(ii)* accepting without strong verification risks a type-I mistake, which under calibration occurs with probability $1 - W$ and is weighted by $\lambda_1$ (after normalizing by how often $g = 0$ occurs, via $\alpha_0$); *(iii)* rejecting without strong verification risks a type-II mistake, which under calibration occurs with probability $W$ and is weighted by $\lambda_2$ (normalized by $\alpha_1$). Thus the optimal policy, pointwise in $W$, chooses the cheapest of these three options, and the optimal value is the expected cost of that choice.

This expression makes clear that, beyond calibration, the *sharpness* of the weak verifier plays a crucial role. Here, sharpness refers to how often $W$ takes values near 0 or 1, i.e., confidently accepts or rejects. If $W$ is often close to 0, then rejection is cheap because the type-II risk $W$ is small; if $W$ is often close to 1, then acceptance is cheap because the type-I risk $1 - W$ is small. In both cases, the policy can avoid strong verification while keeping errors small. In contrast, if $W$ is typically ambiguous (e.g., concentrated near $1/2$), then both risks $W$ and $1 - W$ are non-negligible, so the minimum is typically close to 1, forcing the policy to rely more frequently on strong verification.

> **Takeaway.** Two complementary properties govern the usefulness of a weak verifier: *calibration* makes its scores interpretable as correctness probabilities, while *sharpness* makes it effective by producing decisive scores near 0 or 1.

This message may be useful for practitioners designing and evaluating scalable weak verifiers, although we do not revisit it further in this paper. In contrast, the threshold structure of Theorem 4.2 will serve as a key structural cornerstone for the algorithm developed in the next section.

## 5. Selective Strong Verification and Guarantees

In this section, we design Selective Strong Verification (SSV), an algorithm for the sequential problem introduced in Section 3, fixing a language model $f$, a weak verifier $w$, and a strong verifier $g$. Building on the two-threshold structure identified in Section 4, we construct a policy that adaptively decides, based on the weak score $w_t$, whether to accept, reject, or query strong verification. We make no assumptions on the query stream, the verification signals, or the behavior of the language model. Our goal is to control the empirical type-I and type-II errors $\mathrm{Err}_{\mathrm{I}}(T)$ and $\mathrm{Err}_{\mathrm{II}}(T)$, defined as in Section 3, by enforcing these constraints *uniformly over time*, in the sense that for every $T$ and user-defined $\alpha$ and $\beta$,

$$\mathrm{Err}_{\mathrm{I}}(T) \leq \alpha \quad \text{and} \quad \mathrm{Err}_{\mathrm{II}}(T) \leq \beta.$$

This goal is achieved by the end of this section: Theorem 5.1 establishes that the proposed algorithm enjoys distribution-

---

**Algorithm 1** Selective Strong Verification (SSV)

---

**Require:** Targets $\alpha, \beta \in (0,1)$; exploration probabilities $\{q_A^t, q_R^t\}_{t \geq 1}$; step sizes $\{\eta_t\}_{t \geq 1}$; initial thresholds $\tau_R^1 \leq \tau_A^1$
  **for** $t = 1, 2, \ldots$ **do**
    Receive query and response $(P_t, R_t)$
    $w_t \leftarrow w(P_t, R_t)$ {weak verification score}

———————— *Three regions* ————————
    **if** $w_t > \tau_A^t$ **then**
      $a_t \leftarrow$ A w.p. $1 - q_A^t$, else $a_t \leftarrow$ SV.   Also, set $q_t = q_A^t$.
    **else if** $w_t < \tau_R^t$ **then**
      $a_t \leftarrow$ R w.p. $1 - q_R^t$, else $a_t \leftarrow$ SV.   Also, set $q_t = q_R^t$.
    **else**
      $a_t \leftarrow$ SV.   Also, set $q_t = 1$.
    **end if**

———————— *Final decision and feedback* ————————
    **if** $a_t =$ A **then**
      **accept** $R_t$; **continue**
    **else if** $a_t =$ R **then**
      **reject** $R_t$; **continue**
    **else**
      Observe $g_t \leftarrow g(P_t, R_t)$ {strong verification}
      **if** $g_t = 1$ **then**
        **accept** $R_t$
      **else**
        **reject** $R_t$
      **end if**

———————— *Threshold updates* ————————
$$\tau_A^{t+1} \leftarrow \max\left\{\tau_R^t,\ \tau_A^t + \eta_t \frac{\mathbf{1}\{g_t=0\}\left(\mathbf{1}\{w_t > \tau_A^t\} - \alpha\right)}{q_t}\right\}$$
$$\tau_R^{t+1} \leftarrow \min\left\{\tau_A^{t+1},\ \tau_R^t + \eta_t \frac{\mathbf{1}\{g_t=1\}\left(\beta - \mathbf{1}\{w_t < \tau_R^t\}\right)}{q_t}\right\}$$
    **end if**
    **if** $a_t \neq$ SV **then**
      $\tau_R^{t+1} \leftarrow \tau_R^t, \tau_A^{t+1} \leftarrow \tau_A^t$
    **end if**
  **end for**

---

free, uniform-in-time control of type-I and type-II errors up to finite-sample slack terms.

**Algorithm design.** Algorithm 1 induces a policy via two adaptive thresholds $(\tau_R^t, \tau_A^t)$ and exploration probabilities $(q_A^t, q_R^t)$. Looking at the *Three regions* step of Algorithm 1, at time $t$, the weak score $w_t$ places the response into one of three regions: an *accept region* ($w_t > \tau_A^t$), a *reject region* ($w_t < \tau_R^t$), or an *uncertain region* (between the thresholds). In the uncertain region, the policy always queries the strong verifier. In the accept and reject regions, the response is accepted or rejected, without observing the strong signal $g_t$, with high probability (think of $(q_A^t, q_R^t)$ as small). With a small probability, the policy instead queries the strong verifier and follows its outcome. We explain the necessity of this randomized exploration later.

**Threshold updates.** As is clear from the *Threshold update* step of Algorithm 1, the thresholds are updated only at rounds where the strong signal $g_t$ is observed. Consider the

update of the accept threshold,

$$\tau_A^{t+1} \leftarrow \max\left\{\tau_R^t, \tau_A^t + \eta_t \frac{\mathbf{1}\{g_t = 0\}(\mathbf{1}\{w_t > \tau_A^t\} - \alpha)}{q_t}\right\}.$$

The outer $\max\{\cdot\}$ acts as a projection step that enforces the ordering constraint $\tau_R^t \leq \tau_A^t$ at all times and prevents the thresholds from crossing. The second term inside the maximum is responsible for error tracking: since $\tau_A^t$ controls incorrect acceptances, the update is active only when $g_t = 0$, corresponding to a potential type-I error. The term $\mathbf{1}\{w_t > \tau_A^t\} - \alpha$ drives the empirical rate of accepting among such rounds toward the target level $\alpha$. Finally, the division by $q_t$ corrects for the fact that strong feedback is observed at random times with probabilities that depend on the weak signal, yielding an unbiased update via standard importance weighting. The update for $\tau_R^{t+1}$ is defined analogously to control type-II error.

**Why randomized strong verification is necessary.** The error quantities above are defined *conditionally on the strong verifier*. For example, the type-I error measures the rate of acceptances among times when $g_t = 0$. If the policy deterministically accepts or rejects without querying the strong verifier, then $g_t$ remains unobserved, and the algorithm receives no direct feedback about whether that decision contributed to type-I or type-II error. This way, the interaction may enter a regime where (for instance) $w_t > \tau_A^t$ almost always, leading to many acceptances but essentially no strong-verification feedback. In such regimes, violations of the error constraints cannot be detected or corrected without additional assumptions. Algorithm 1 avoids this issue by querying the strong verifier with small probabilities $q_A^t$ and $q_R^t$ even in decisive regions.

The following theorem shows that Algorithm 1 controls the empirical errors for any $T$ (up to finite-sample slack terms), without any assumptions on the stream $\{(P_t, R_t)\}_{t \geq 1}$.

**Theorem 5.1** (Finite-time empirical error control). *Fix a horizon $T \geq 1$. Run Algorithm 1 with a constant step size $\eta_t = \eta > 0$ ($\forall, t$) and predictable strong-query probabilities $\{q_t\}_{t=1}^T$ satisfying $q_t \in (0, 1]$ and $q_{\min} := \min_{t \leq T}\{q_A^t, q_R^t\} > 0$. Assume the initial thresholds satisfy $\tau_R^1 \leq \tau_A^1$ with $\tau_R^1, \tau_A^1 \in [0, 1]$. Then for any $\delta \in (0, 1)$, with probability at least $1 - \delta$ over the algorithm's internal randomness,*

$$\mathrm{Err}_{\mathrm{I}}(T) \leq \alpha + \Delta(N_0(T), \delta),$$
$$\mathrm{Err}_{\mathrm{II}}(T) \leq \beta + \Delta(N_1(T), \delta),$$

*where*

$$\Delta(N, \delta) := \frac{1 + \frac{2\eta}{q_{\min}}}{\eta N} + \sqrt{\frac{2\log(4/\delta)}{N q_{\min}}} + \frac{\log(4/\delta)}{3 N q_{\min}},$$

*with the convention $\Delta(0, \delta) = 0$.*

The error bound decomposes into two qualitatively different sources. The first term, of order $1/(\eta N)$, reflects the intrinsic error of tracking a target quantile with an online threshold update: even in an idealized setting where the strong signal were always revealed after each decision, this term would remain. It is the same fundamental limitation that appears in online conformal prediction and other quantile-tracking procedures (Gibbs & Candès, 2021; Angelopoulos et al., 2023; Ramalingam et al., 2025). The remaining two terms arise from partial access to the strong signal and the randomized exploration required to obtain unbiased feedback. Their dependence on $q_{\min}$ is insightful: larger exploration probabilities make error control easier, but increase the frequency of strong verification calls. This highlights an inherent algorithmic trade-off induced by our design—balancing statistical accuracy against verification cost—and suggests that, in practice, the exploration probabilities should be treated as tunable hyperparameters to achieve the best performance.

## 6. Experimental Results

In this section, we empirically evaluate SSV (Algorithm 1) across two distinct reasoning paradigms, demonstrating the generality of the problem formulation in Section 3.

The first scenario considers outcome-level verification, consistent with the Outcome Reward Modeling (ORM) paradigm (Cobbe et al., 2021). Here, the system evaluates a complete candidate solution after it has been fully generated. We utilize the MATH dataset (Hendrycks et al., 2021), a rigorous benchmark for mathematical problem-solving. Following the notation in Section 3, a query $P_t$ represents a user prompt, and $R_t$ is a complete response candidate. The model generates responses sequentially; the policy $\pi_t$ evaluates each via its weak score $w_t$ until a candidate is accepted or the budget $n$ is exhausted. To test SSV across varying complexities, we conduct experiments on problems from difficulty levels 2, 3, and 5.

The second scenario focuses on step-by-step verification, mirroring Process Reward Modeling (PRM) (Lightman et al., 2023). We evaluate this using Sudoku puzzles (Shahab, 2023), where the focus shifts to granular, sequential decision-making. In this task, $P_t$ represents the current board state (initial puzzle plus all accepted digits), and $R_t$ is the model's proposed next digit and its coordinates. Sudoku is a high-stakes environment where a single incorrect step often renders the entire puzzle unsolvable, testing the policy's ability to intercept errors at the step level while minimizing calls to the costly strong verifier.

We provide specific implementation details for the weak and strong verifiers used in each task in Appendix B.6.

Our goal is to show that across both tasks, SSV (Algo-

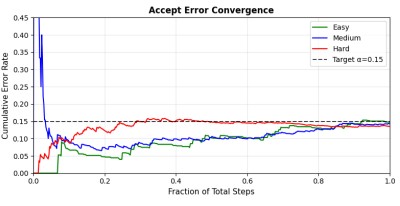
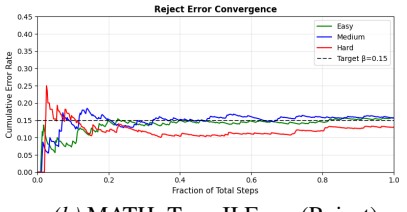
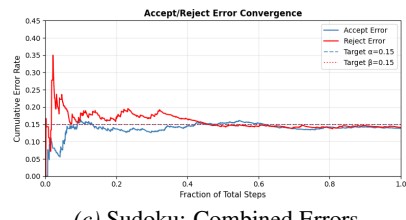

*(a)* MATH: Type-I Error (Accept)   *(b)* MATH: Type-II Error (Reject)   *(c)* Sudoku: Combined Errors

*Figure 2.* **Empirical Error Rate Convergence.** Running-average error rates $\frac{1}{T}\sum_{t=1}^{T}\mathrm{err}_t$ for target levels $\alpha = \beta = 0.15$. **Left and Center (MATH):** Convergence of Type-I and Type-II errors across three difficulty levels in the outcome level verification task. **Right (Sudoku):** Convergence in the sequential step-by-step reasoning task.

rithm 1) successfully controls empirical Type-I and Type-II errors at their nominal levels. Furthermore, we show that our approach allows a service provider to principledly interpolate between two operational extremes. On one end is the *Weak-Only regime*, where the system relies solely on the weak verifier. This represents the absolute lower bound for computation overhead and latency caused by using a strong verification as it bypasses strong verifier entirely; however this efficiency comes at the cost of reasoning accuracy and user trust. On the opposite end is the *Strong-Only regime*, where every query is routed to the strong verifier. while this maximizes reasoning performance and reliability, it incurs the highest possible operational costs and latency. We demonstrate that SSV can identify a favorable balance, maintaining high reasoning performance while significantly reducing the load on the strong verifier.

**Experimental Setup and Metrics.** We report results along two primary axes. First, we measure empirical Type-I and Type-II error rates (Section 3) to validate that our algorithm controls these quantities under non-stationary conditions. Second, we assess the reasoning performance vs. verification-cost (measured by the strong-verification frequency $\mathrm{SV}(T)$) trade-off by sweeping the target error levels $(\alpha, \beta)$.

**Baselines.** We compare our policy against two baselines that represent the theoretical and practical boundaries of the system. The **Strong-Only (Oracle)** baseline invokes the strong verifier for every query, representing the maximum achievable reasoning accuracy and the upper bound for operational cost. Conversely, the **Weak-Only (Greedy)** baseline represents the minimum-cost regime; here, the system generates $n$ candidates and selects the one with the highest weak verification score to be accepted as the final result or the next reasoning step, bypassing the strong verifier entirely.

### 6.1. Empirical Error Control

We first evaluate the ability of SSV (Algorithm 1) to maintain target error levels. Figure 2 displays the *running average* of the Type-I and Type-II error rates as the interaction

sequence progresses, defined at time step $T$ as $\frac{1}{T}\sum_{t=1}^{T}\mathrm{err}_t$. Here, the index $t$ iterates over the global stream of verification decisions across all problems in each dataset.

Across both the mathematical reasoning tasks (Fig. 2a, b) and the sequential Sudoku steps (Fig. 2c), the running average errors stabilize near the nominal targets of $\alpha = \beta = 0.15$. This confirms the result of Theorem 5.1. The trajectories reflect the typical behavior of online quantile tracking, where the learning rate $\eta$ governs the trade-off between adaptation speed and stability.[2]

### 6.2. Reasoning Performance vs. Verification Cost

To characterize our framework's operational potential, we sweep the error targets $(\alpha, \beta)$ and plot reasoning accuracy against the average number of strong-verifier calls per problem. This visualization reveals the Pareto frontier of the accuracy-verification-cost trade-off, spanning the zero-cost Weak-Only baseline to the high-reliability Strong-Only Oracle. SSV serves as a principled mechanism to navigate and realize this frontier, allowing service providers to interpolate between these regimes by desired reliability bounds.

The intrinsic efficiency of the Pareto curve (as reflected by its slope, that is, how much accuracy is gained per unit of strong verification cost) is fundamentally governed by the weak verifier signals. The algorithm's role is then to leverage this sharpness by dynamically identifying which reasoning steps require strong verification. In the Easy MATH subsets and the Sudoku task, the weak signals exhibit high sharpness, resulting in steeper curves. In these cases, the inherent quality of the signals enables near-Oracle accuracy with only a fraction of the strong verifier calls. For instance, in Sudoku, the Strong-Only Oracle reaches an accuracy of 44.2% with 5.32 strong calls per puzzle. In contrast, our adaptive policy achieves a comparable 43.1% accuracy (at $\alpha = \beta = 0.01$) while requiring only 2.87 strong calls—a 46% reduction in the load on the strong verifier.

---

[2]Larger values of $\eta$ facilitate faster initial convergence but result in noisier trajectories, while smaller values produce smoother curves with slower convergence.

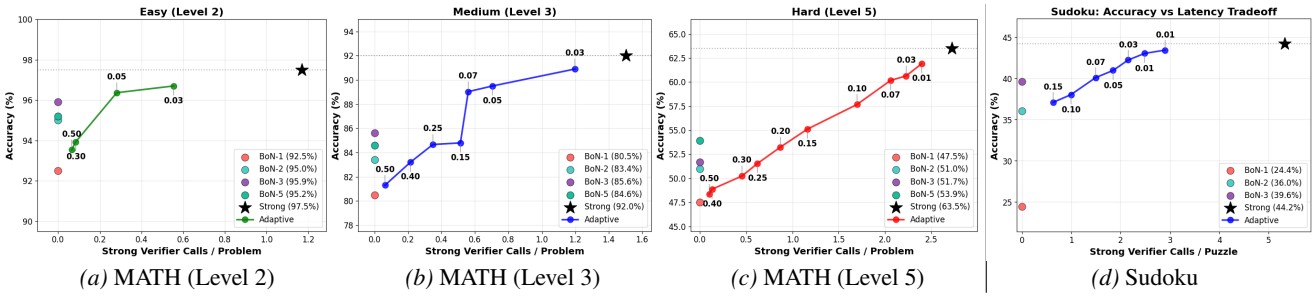

*(a)* MATH (Level 2)     *(b)* MATH (Level 3)     *(c)* MATH (Level 5)     *(d)* Sudoku

*Figure 3.* **Reasoning Accuracy vs. Verification Cost Tradeoffs.** SSV (Algorithm 1) (Adaptive: solid colored lines) interpolates between the **Strong-Only Oracle** (black star) and the **Weak-Only baselines** (colored circles). **Left (MATH):** Tradeoff curves for Easy, Medium, and Hard problems; **Right (Sudoku):** Tradeoff curve for Sudoku step-by-step reasoning task. points are labeled with nominal error targets where $\alpha = \beta$.

In the hardest MATH setting (level 5), the relationship is more linear: as the weak signals become less decisive, higher accuracy requires an approximately proportional increase in strong verification. Yet SSV still concentrates resources where they are most needed, reaching 60% accuracy with 2 calls per problem, compared to the Oracle's 2.8 calls for 63.5% accuracy. For a detailed analysis of the weak verification scores, refer to Appendix B.3.

Finally, we observe a compounded verification efficiency in the sequential reasoning setting. As shown in the last column of Table 1, SSV is more query-efficient with the weak verifier than the Weak-Only baseline. While the Weak-Only baseline requires 6.00 weak calls per puzzle to reach its lower success rate, our policy averages between 4.8–5.2 weak calls across all operating points. This suggests that by explicitly modeling uncertainty through adaptive thresholds, SSV avoids redundant reasoning steps. It achieves this in two ways: by accepting a confident result early or by escalating immediately to the strong verifier when the weak signal is deemed insufficient to support the target reliability. This immediate escalation bypasses further weak verification cycles that are statistically likely to be uninformative.

*Table 1.* **Performance Tradeoffs on Sudoku.** Detailed comparison across different error regimes.

| Method | Accuracy | Strong/Puzzle | Weak/Puzzle |
|---|---|---|---|
| Strong Baseline (Oracle) | 44.2% | 5.32 | – |
| SSV ($\alpha = \beta = 0.001$) | 43.1% | 3.31 | 5.22 |
| SSV ($\alpha = \beta = 0.01$) | 43.1% | 2.87 | 5.19 |
| SSV ($\alpha = \beta = 0.03$) | 42.7% | 2.14 | 5.20 |
| SSV ($\alpha = \beta = 0.05$) | 42.1% | 2.06 | 5.22 |
| SSV ($\alpha = \beta = 0.10$) | 39.0% | 1.24 | 5.18 |
| SSV ($\alpha = \beta = 0.20$) | 36.2% | 0.76 | 5.04 |
| SSV ($\alpha = \beta = 0.30$) | 34.5% | 0.68 | 4.84 |
| Weak Baseline (Greedy) | 33.6% | 0.00 | 6.00 |

**Summary of Results.** Across both mathematical reasoning and sequential puzzle-solving, our experiments yield three key findings. First, SSV consistently maintains target error rates without prior knowledge of the query distribution or verifier quality. Second, our framework provides a smooth Pareto frontier that allows users to prioritize either reasoning quality or operational cost. Finally, by identifying and acting upon the "sharpness" of the weak signal, SSV achieves near-oracle performance with a fraction of the expensive verification load when possible.

## 7. Discussion and Future Work

We introduced a principled algorithmic framework for orchestrating weak and strong verification in LLM reasoning, showing–both theoretically and empirically–that it is possible to achieve reasoning performance comparable to always applying strong verification while querying the strong verifier only a small fraction of the time. A key limitation of the current framework is that the decision to use strong verification depends only on the weak score, and not on the broader prompt–response context $(P_t, R_t)$; consequently, our guarantees control type-I and type-II errors only in a marginal sense, averaged over all rounds. We view this as a modeling choice rather than a fundamental limitation. Incorporating contextual dependence into weak–strong verification policies could enable finer-grained and more efficient allocation of strong verification, but requires more complex online calibration procedures, including context-dependent thresholds and conditional error control under partial feedback. We leave this as an important direction for future work.

## Acknowledgements

The authors thank EnCORE, the Institute for Emerging CORE Methods in Data Science, for their support. SK additionally acknowledges support from a gift from AWS to Penn Engineering's ASSET Center for Trustworthy AI.

## Impact Statement

This work provides a principled framework for allocating costly verification resources in reasoning systems, enabling reliable use of large language models at scale. By reducing unnecessary human and computational verification while preserving correctness guarantees, our approach can lower deployment costs and improve the safety and trustworthiness of AI-assisted decision making across high-stakes domains.

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

## A. Proofs

**Proof of Theorem 4.2 and Proposition 4.3:**

*Proof.* Let $(P, R)$ be distributed as $P \sim \mathcal{D}$ and $R = f(P)$. Define the random variables

$$W := w(P, R) \in [0, 1], \qquad g := g(P, R) \in \{0, 1\},$$

and let $\alpha_0 := \Pr[g = 0]$ and $\alpha_1 := \Pr[g = 1]$. Assume $\alpha_0, \alpha_1 > 0$ (otherwise the conditional error terms in (1) are vacuous and the problem is trivial). For $\lambda_1, \lambda_2 \geq 0$, consider the population objective

$$V(\lambda_1, \lambda_2) := \min_{\pi:[0,1] \to \{A,R,SV\}} \Pr[\pi(W) = SV] + \lambda_1 \Pr[\pi(W) = A \mid g = 0] + \lambda_2 \Pr[\pi(W) = R \mid g = 1].$$

Define the effective weights

$$a := \frac{\lambda_1}{\alpha_0}, \qquad b := \frac{\lambda_2}{\alpha_1}.$$

**Claim 1** (Unconditional and pointwise form). *Under Assumption 3.1 (calibration), the objective can be written as*

$$V(\lambda_1, \lambda_2) = \min_{\pi:[0,1] \to \{A,R,SV\}} \mathbb{E}\left[\ell\left(W, \pi(W)\right)\right],$$

*where the pointwise action costs are*

$$\ell(w, SV) = 1, \qquad \ell(w, A) = a(1 - w), \qquad \ell(w, R) = bw.$$

*Moreover, an optimal policy can be chosen deterministically and pointwise:*

$$\pi^\star(w) \in \arg\min_{u \in \{A,R,SV\}} \ell(w, u) \quad \text{for all } w \in [0, 1],$$

*and hence*

$$V(\lambda_1, \lambda_2) = \mathbb{E}[\min\{1, a(1 - W), bW\}].$$

*Proof.* First rewrite the conditional terms:

$$\Pr[\pi(W) = A \mid g = 0] = \frac{\Pr[\pi(W) = A, g = 0]}{\alpha_0}, \qquad \Pr[\pi(W) = R \mid g = 1] = \frac{\Pr[\pi(W) = R, g = 1]}{\alpha_1}.$$

Thus,

$$V(\lambda_1, \lambda_2) = \min_\pi \left( \Pr[\pi(W) = SV] + a \Pr[\pi(W) = A, g = 0] + b \Pr[\pi(W) = R, g = 1] \right).$$

Conditioning on $W$ and using the law of total expectation,

$$\Pr[\pi(W) = A, g = 0] = \mathbb{E}[\mathbf{1}\{\pi(W) = A\} \Pr(g = 0 \mid W)],$$
$$\Pr[\pi(W) = R, g = 1] = \mathbb{E}[\mathbf{1}\{\pi(W) = R\} \Pr(g = 1 \mid W)].$$

By Assumption 3.1, $\Pr(g = 1 \mid W) = W$ and $\Pr(g = 0 \mid W) = 1 - W$, so

$$V(\lambda_1, \lambda_2) = \min_\pi \mathbb{E}[\mathbf{1}\{\pi(W) = SV\} + a \mathbf{1}\{\pi(W) = A\}(1 - W) + b \mathbf{1}\{\pi(W) = R\}W] = \min_\pi \mathbb{E}\left[\ell\left(W, \pi(W)\right)\right],$$

with $\ell$ as stated.

For the pointwise optimality: for each $w$, the quantity inside the expectation depends on $\pi$ only through the chosen action at $w$. If $\pi$ is randomized, then conditional on $W = w$ its expected cost is a convex combination of $\{\ell(w, A), \ell(w, R), \ell(w, SV)\}$, which is minimized by placing all mass on an action that attains the minimum. Thus an optimal policy exists that is deterministic and satisfies $\pi^\star(w) \in \arg\min_u \ell(w, u)$ for all $w$, and the optimal value is

$$V(\lambda_1, \lambda_2) = \mathbb{E}[\min\{1, a(1 - W), bW\}].$$

$\square$

The value identity in the claim is exactly Proposition 3.3.

It remains to show the threshold structure of an optimal policy (Theorem 3.2) by comparing the three affine costs. If $a = 0$, then $\ell(w, A) \equiv 0$ is pointwise minimal and $\pi^\star(w) \equiv A$. If $b = 0$, then $\ell(w, R) \equiv 0$ is pointwise minimal and $\pi^\star(w) \equiv R$. Assume henceforth $a > 0$ and $b > 0$, and define

$$t_{\text{low}} := \frac{1}{b}, \qquad t_{\text{high}} := 1 - \frac{1}{a}.$$

Observe that $\ell(w, SV) \leq \ell(w, R)$ iff $1 \leq bw$ iff $w \geq t_{\text{low}}$, and $\ell(w, SV) \leq \ell(w, A)$ iff $1 \leq a(1 - w)$ iff $w \leq t_{\text{high}}$. Therefore, whenever $t_{\text{low}} \leq t_{\text{high}}$, we have for every $w \in [t_{\text{low}}, t_{\text{high}}]$ that

$$\ell(w, SV) = 1 \leq bw = \ell(w, R) \quad \text{and} \quad \ell(w, SV) = 1 \leq a(1 - w) = \ell(w, A),$$

so $SV$ is pointwise optimal on $[t_{\text{low}}, t_{\text{high}}]$.

For $w < t_{\text{low}}$, we have $bw < 1$, and also $w < t_{\text{low}} \leq t_{\text{high}}$ implies $w < 1 - \frac{1}{a}$, i.e. $a(1 - w) > 1$. Hence $\ell(w, R) = bw < 1 = \ell(w, SV) < a(1 - w) = \ell(w, A)$, so $R$ is pointwise optimal. Similarly, for $w > t_{\text{high}}$, we have $a(1 - w) < 1$, and $w > t_{\text{high}} \geq t_{\text{low}}$ implies $bw > 1$, so $\ell(w, A) < \ell(w, SV) \leq \ell(w, R)$ and $A$ is pointwise optimal. Thus, when $t_{\text{low}} \leq t_{\text{high}}$, one optimal policy is

$$\pi^\star(w) = \begin{cases} R, & w < t_{\text{low}}, \\ SV, & t_{\text{low}} \leq w \leq t_{\text{high}}, \\ A, & w > t_{\text{high}}, \end{cases}$$

with arbitrary tie-breaking at the thresholds, establishing the two-threshold structure in Theorem 3.2.

Finally, in the three-region regime $t_{\text{low}} \leq t_{\text{high}}$, the value formula can be expanded by splitting the event $\{W < t_{\text{low}}\}$, $\{t_{\text{low}} \leq W \leq t_{\text{high}}\}$, and $\{W > t_{\text{high}}\}$:

$$V(\lambda_1, \lambda_2) = b\, \mathbb{E}[W\, \mathbf{1}\{W < t_{\text{low}}\}] + \Pr(t_{\text{low}} \leq W \leq t_{\text{high}}) + a\, \mathbb{E}[(1 - W)\, \mathbf{1}\{W > t_{\text{high}}\}]\,.$$

This completes the proof of both results. $\qquad\square$

**Proof of Theorem 5.1:**

*Proof.* We prove the type-I and type-II bounds simultaneously via a common template. The key idea is that the threshold updates track (via importance weighting) the deviation between a *threshold-induced* error rate and its target level, and the *policy* errors are pointwise dominated by these threshold-induced rates. We control the tracking error with a martingale concentration bound (Freedman), and control the drift of the thresholds by a deterministic bound.

**Notation and setup.** Let $\mathcal{F}_t$ denote the sigma-field generated by the full interaction history up to time $t$ (including $w_{1:t}$, the algorithm's past actions, and all queried strong labels). Let

$$O_t := \mathbf{1}\{a_t = SV\} \in \{0, 1\} \quad \text{and} \quad q_t := \mathbb{P}(O_t = 1 \mid \mathcal{F}_{t-1}, P_t, R_t),$$

so that $q_t$ is predictable and $q_{\min} := \min_{t \leq T} q_t > 0$ by assumption.

Define the latent counts

$$N_0(T) := \sum_{t=1}^{T} \mathbf{1}\{g_t = 0\}, \qquad N_1(T) := \sum_{t=1}^{T} \mathbf{1}\{g_t = 1\}.$$

It will be convenient to define the following *threshold-induced* error rates:

$$\overline{\text{Err}}_{\text{I}}(T) := \frac{1}{N_0(T)} \sum_{t=1}^{T} \mathbf{1}\{g_t = 0\}\, \mathbf{1}\{w_t > \tau_A^t\},$$

$$\overline{\text{Err}}_{\text{II}}(T) := \frac{1}{N_1(T)} \sum_{t=1}^{T} \mathbf{1}\{g_t = 1\}\, \mathbf{1}\{w_t < \tau_R^t\},$$

with the convention that each ratio is $0$ when its denominator is $0$. These differ from the *policy* errors in the theorem,

$$\text{Err}_{\text{I}}(T) = \frac{1}{N_0(T)} \sum_{t=1}^{T} \mathbf{1}\{g_t = 0,\ \pi_t(w_t) = A\}, \qquad \text{Err}_{\text{II}}(T) = \frac{1}{N_1(T)} \sum_{t=1}^{T} \mathbf{1}\{g_t = 1,\ \pi_t(w_t) = R\},$$

because the policy may choose SV (exploration) even when $w_t$ lies in the A/R regions.

For the accept side, define

$$x_t^A := \mathbf{1}\{w_t > \tau_A^t\} - \alpha \in [-1, 1], \qquad e_t^A := \mathbf{1}\{g_t = 0\}\, x_t^A, \qquad \widehat{e}_t^A := \mathbf{1}\{g_t = 0\} \frac{O_t}{q_t}\, x_t^A.$$

For the reject side, define

$$x_t^R := \mathbf{1}\{w_t < \tau_R^t\} - \beta \in [-1, 1], \qquad e_t^R := \mathbf{1}\{g_t = 1\}\, x_t^R, \qquad \widehat{e}_t^R := \mathbf{1}\{g_t = 1\} \frac{O_t}{q_t}\, x_t^R.$$

Finally, define

$$\varepsilon(N, \delta) := \sqrt{\frac{2\log(4/\delta)}{N q_{\min}}} + \frac{\log(4/\delta)}{3N q_{\min}}, \qquad \varepsilon(0, \delta) := 0, \qquad B := 1 + \frac{2\eta}{q_{\min}}.$$

Note that $\Delta(N, \delta) = \frac{B}{\eta N} + \varepsilon(N, \delta)$ (with $\Delta(0, \delta) = 0$).

**Claim 2** (Unbiased importance weighting). *For each $t$,*

$$\mathbb{E}\big[\widehat{e}_t^A \mid \mathcal{F}_{t-1}, P_t, R_t, g_t\big] = e_t^A, \qquad \mathbb{E}\big[\widehat{e}_t^R \mid \mathcal{F}_{t-1}, P_t, R_t, g_t\big] = e_t^R.$$

*Consequently, $Z_t^A := \widehat{e}_t^A - e_t^A$ and $Z_t^R := \widehat{e}_t^R - e_t^R$ are martingale differences.*

*Proof.* We prove the accept statement; the reject statement is identical. Condition on $(\mathcal{F}_{t-1}, P_t, R_t, g_t)$. Then $\mathbf{1}\{g_t = 0\}$ and $x_t^A$ are fixed, while $O_t \sim \text{Bernoulli}(q_t)$ with conditional mean $q_t$. Hence

$$\mathbb{E}\big[\widehat{e}_t^A \mid \mathcal{F}_{t-1}, P_t, R_t, g_t\big] = \mathbf{1}\{g_t = 0\}\, x_t^A\, \mathbb{E}\bigg[\frac{O_t}{q_t} \mid \mathcal{F}_{t-1}, P_t, R_t, g_t\bigg] = \mathbf{1}\{g_t = 0\}\, x_t^A = e_t^A.$$

Therefore $\mathbb{E}[Z_t^A \mid \mathcal{F}_{t-1}, P_t, R_t, g_t] = 0$, i.e., $\{Z_t^A\}$ is a martingale-difference sequence. $\square$

**Claim 3** (Telescoping inequalities from the threshold updates). *For every $T \geq 1$,*

$$\sum_{t=1}^{T} \widehat{e}_t^A \leq \frac{\tau_A^{T+1} - \tau_A^1}{\eta}, \qquad \sum_{t=1}^{T} \widehat{e}_t^R \leq \frac{\tau_R^1 - \tau_R^{T+1}}{\eta}.$$

*Proof.* For the accept threshold, Algorithm 1 updates (in terms of $O_t$)

$$\tau_A^{t+1} = \max\{\tau_R^t,\ \tau_A^t + \eta \widehat{e}_t^A\} \geq \tau_A^t + \eta \widehat{e}_t^A,$$

so $\widehat{e}_t^A \leq (\tau_A^{t+1} - \tau_A^t)/\eta$, and summing yields $\sum_{t \leq T} \widehat{e}_t^A \leq (\tau_A^{T+1} - \tau_A^1)/\eta$.

For the reject threshold, Algorithm 1 updates

$$\tau_R^{t+1} = \min\{\tau_A^{t+1},\ \tau_R^t - \eta \widehat{e}_t^R\} \leq \tau_R^t - \eta \widehat{e}_t^R,$$

so $\widehat{e}_t^R \leq (\tau_R^t - \tau_R^{t+1})/\eta$, and summing yields $\sum_{t \leq T} \widehat{e}_t^R \leq (\tau_R^1 - \tau_R^{T+1})/\eta$. $\square$

**Claim 4** (Uniform boundedness of thresholds). *For all $t \leq T + 1$,*

$$\tau_A^t, \tau_R^t \in \bigg[ -\frac{\eta}{q_{\min}},\ 1 + \frac{\eta}{q_{\min}} \bigg],$$

*and hence*

$$|\tau_A^{T+1} - \tau_A^1| \leq B, \qquad |\tau_R^{T+1} - \tau_R^1| \leq B.$$

*Proof.* We prove the bound for $\tau_A^t$; the argument for $\tau_R^t$ is analogous.

First, note that $|x_t^A| \leq 1$ and $O_t/q_t \leq 1/q_{\min}$ imply

$$|\widehat{e}_t^A| = \mathbf{1}\{g_t = 0\} \frac{O_t}{q_t} |x_t^A| \leq \frac{1}{q_{\min}}.$$

Thus whenever $\tau_A$ changes at all, its *raw* increment is bounded: $|\eta\,\widehat{e}_t^A| \leq \eta/q_{\min}$.

Now use the fact that $w_t \in [0, 1]$. If $\tau_A^t \geq 1$, then $\mathbf{1}\{w_t > \tau_A^t\} = 0$, so $x_t^A = -\alpha \leq 0$ and any raw update can only decrease $\tau_A$. If $\tau_A^t \leq 0$, then $\mathbf{1}\{w_t > \tau_A^t\} = 1$, so $x_t^A = 1 - \alpha \geq 0$ and any raw update can only increase $\tau_A$. Therefore, starting from $\tau_A^1 \in [0, 1]$, the process can overshoot the interval $[0, 1]$ by at most one raw step, i.e., by at most $\eta/q_{\min}$, which yields

$$\tau_A^t \in \left[ -\frac{\eta}{q_{\min}}, \; 1 + \frac{\eta}{q_{\min}} \right] \quad \text{for all } t \leq T + 1.$$

The bound $|\tau_A^{T+1} - \tau_A^1| \leq 1 + 2\eta/q_{\min} = B$ follows since $\tau_A^1 \in [0, 1]$. The same reasoning applies to $\tau_R^t$. $\qquad\square$

**Claim 5** (Freedman bound; variance computed explicitly). *Let*

$$M_T^A := \sum_{t=1}^{T} (\widehat{e}_t^A - e_t^A), \qquad M_T^R := \sum_{t=1}^{T} (\widehat{e}_t^R - e_t^R).$$

*Then, with probability at least $1 - \delta/2$,*

$$|M_T^A| \leq N_0(T)\,\varepsilon(N_0(T), \delta), \qquad |M_T^R| \leq N_1(T)\,\varepsilon(N_1(T), \delta),$$

*with the convention that the right-hand side is 0 when the corresponding $N_i(T) = 0$.*

*Proof.* We prove the accept statement; the reject statement is identical.

Define $Z_t^A := \widehat{e}_t^A - e_t^A = \mathbf{1}\{g_t = 0\}\, x_t^A \left(\frac{O_t}{q_t} - 1\right)$. Then $M_T^A = \sum_{t \leq T} Z_t^A$, and by Claim 2 it is a martingale.

*Bounded increments.* Since $|x_t^A| \leq 1$ and $q_t \geq q_{\min}$,

$$|Z_t^A| = \mathbf{1}\{g_t = 0\}\, |x_t^A| \cdot \left|\frac{O_t}{q_t} - 1\right| \leq \left|\frac{O_t}{q_t} - 1\right| \leq \max\left\{1, \frac{1}{q_t} - 1\right\} \leq \frac{1}{q_t} \leq \frac{1}{q_{\min}}.$$

*Conditional variance (explicit computation).* Because $\mathbb{E}[Z_t^A \mid \mathcal{F}_{t-1}, P_t, R_t, g_t] = 0$,

$$\mathrm{Var}(Z_t^A \mid \mathcal{F}_{t-1}, P_t, R_t, g_t) = \mathbb{E}\left[(Z_t^A)^2 \mid \mathcal{F}_{t-1}, P_t, R_t, g_t\right].$$

Now

$$(Z_t^A)^2 = \mathbf{1}\{g_t = 0\}\,(x_t^A)^2 \left(\frac{O_t}{q_t} - 1\right)^2.$$

Conditioning on $(\mathcal{F}_{t-1}, P_t, R_t, g_t)$ makes $\mathbf{1}\{g_t = 0\}(x_t^A)^2$ fixed, so it remains to compute

$$\mathbb{E}\left[\left(\frac{O_t}{q_t} - 1\right)^2 \;\middle|\; \mathcal{F}_{t-1}, P_t, R_t, g_t\right].$$

Since $O_t \sim \text{Bernoulli}(q_t)$ given $(\mathcal{F}_{t-1}, P_t, R_t)$, we have

$$\begin{aligned}
\mathbb{E}\left[\left(\frac{O_t}{q_t} - 1\right)^2\right] &= q_t \left(\frac{1}{q_t} - 1\right)^2 + (1 - q_t)\left(0 - 1\right)^2 \\
&= q_t \cdot \frac{(1 - q_t)^2}{q_t^2} + (1 - q_t) \\
&= \frac{(1 - q_t)^2}{q_t} + (1 - q_t) \\
&= (1 - q_t)\left(\frac{1 - q_t}{q_t} + 1\right) = \frac{1 - q_t}{q_t}.
\end{aligned}$$

Therefore,

$$\text{Var}(Z_t^A \mid \mathcal{F}_{t-1}, P_t, R_t, g_t) = \mathbf{1}\{g_t = 0\} (x_t^A)^2 \frac{1 - q_t}{q_t} \leq \mathbf{1}\{g_t = 0\} \frac{1}{q_t} \leq \frac{\mathbf{1}\{g_t = 0\}}{q_{\min}}.$$

Summing over $t \leq T$ gives the predictable variance proxy

$$V_T^A := \sum_{t=1}^{T} \text{Var}(Z_t^A \mid \mathcal{F}_{t-1}, P_t, R_t, g_t) \leq \frac{N_0(T)}{q_{\min}}.$$

*Apply Freedman's inequality.* Freedman's inequality for martingales with increment bound $b = 1/q_{\min}$ and variance proxy $V_T^A \leq N_0(T)/q_{\min}$ implies that, allocating tail probability $\delta/4$ to each side (two-sided bound),

$$|M_T^A| \leq \sqrt{2 \frac{N_0(T)}{q_{\min}} \log \frac{4}{\delta}} + \frac{1}{3q_{\min}} \log \frac{4}{\delta} = N_0(T) \, \varepsilon(N_0(T), \delta),$$

with the convention $N_0(T)\varepsilon(N_0(T), \delta) = 0$ if $N_0(T) = 0$. $\qquad\square$

**Control the threshold-induced rates.**  Observe that

$$N_0(T)\big(\overline{\text{Err}}_{\text{I}}(T) - \alpha\big) = \sum_{t=1}^{T} \mathbf{1}\{g_t = 0\}\big(\mathbf{1}\{w_t > \tau_A^t\} - \alpha\big) = \sum_{t=1}^{T} e_t^A.$$

Also $\sum_{t=1}^{T} e_t^A = \sum_{t=1}^{T} \widehat{e}_t^A - M_T^A$. Using Claim 3 and Claim 4,

$$\sum_{t=1}^{T} \widehat{e}_t^A \leq \frac{\tau_A^{T+1} - \tau_A^1}{\eta} \leq \frac{|\tau_A^{T+1} - \tau_A^1|}{\eta} \leq \frac{B}{\eta}.$$

Combining with Claim 5 gives, with probability at least $1 - \delta/2$,

$$\sum_{t=1}^{T} e_t^A \leq \frac{B}{\eta} + |M_T^A| \leq \frac{B}{\eta} + N_0(T) \, \varepsilon(N_0(T), \delta).$$

Divide by $N_0(T)$ (or use the $N_0(T) = 0$ convention) to obtain

$$\overline{\text{Err}}_{\text{I}}(T) \leq \alpha + \frac{B}{\eta N_0(T)} + \varepsilon(N_0(T), \delta) = \alpha + \Delta(N_0(T), \delta).$$

The same argument on the reject side yields, with probability at least $1 - \delta/2$,

$$\overline{\text{Err}}_{\text{II}}(T) \leq \beta + \Delta(N_1(T), \delta).$$

**From threshold-induced to policy errors (domination).**  If $\pi_t(w_t) = A$, then the algorithm must be in the accept region, hence $w_t > \tau_A^t$. Therefore,

$$\mathbf{1}\{g_t = 0, \, \pi_t(w_t) = A\} \leq \mathbf{1}\{g_t = 0\}\mathbf{1}\{w_t > \tau_A^t\},$$

and summing and dividing by $N_0(T)$ gives $\text{Err}_{\text{I}}(T) \leq \overline{\text{Err}}_{\text{I}}(T)$. Similarly, if $\pi_t(w_t) = R$ then necessarily $w_t < \tau_R^t$, so $\text{Err}_{\text{II}}(T) \leq \overline{\text{Err}}_{\text{II}}(T)$.

**Finish by a union bound.**  Each of the two events $\{\overline{\text{Err}}_{\text{I}}(T) \leq \alpha + \Delta(N_0(T), \delta)\}$ and $\{\overline{\text{Err}}_{\text{II}}(T) \leq \beta + \Delta(N_1(T), \delta)\}$ holds with probability at least $1 - \delta/2$. By a union bound, both hold with probability at least $1 - \delta$. Using domination then yields, on the same event,

$$\text{Err}_{\text{I}}(T) \leq \alpha + \Delta(N_0(T), \delta), \qquad \text{Err}_{\text{II}}(T) \leq \beta + \Delta(N_1(T), \delta),$$

which proves the theorem. $\qquad\square$

# B. Additional Experiments

## B.1. Additional Error Control Results

In this section, we provide additional evidence of the algorithm's convergence across varying target error configurations $(\alpha, \beta)$. The primary goal is to demonstrate that the online calibration mechanism is effective regardless of the specific reliability requirements set by the service provider.

### B.1.1. OUTCOME-LEVEL VERIFICATION (MATH DATASET)

Figures 4, 5, and 6 show the running error convergence for the MATH dataset across three difficulty levels. These plots include a histogram on the left of each panel, which visualizes the final cumulative running errors at the last iteration. The specific hyperparameters used for these runs, including the learning rates ($\eta$) and initial thresholds ($\tau_0$), are detailed in the accompanying tables.

Tables 2, 3, and 4 detail the specific hyperparameter used—including the learning rates ($\eta$) and initial thresholds ($\tau_0$)—and provide the final empirical error rates (Err A/R) attained at the end of the interaction sequence for reproducibility.

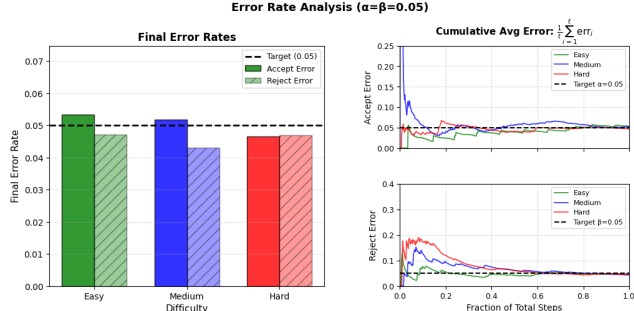

*Figure 4.* **Best-of-$n$ (MATH): error convergence for $\alpha = \beta = 0.05$.**

| Diff. | $\eta_A/\eta_R$ | $\tau_{A,0}/\tau_{R,0}$ | Err (A/R) |
|-------|-----------------|------------------------|-----------|
| Lvl 2 | 0.05 / 0.02     | 0.70 / 0.15            | .053 / .047 |
| Lvl 3 | 0.05 / 0.01     | 0.90 / 0.15            | .052 / .043 |
| Lvl 5 | 0.05 / 0.01     | 0.80 / 0.15            | .047 / .047 |

*Table 2.* Hyperparameters and final errors for Figure 4 ($\alpha = \beta = 0.05$).

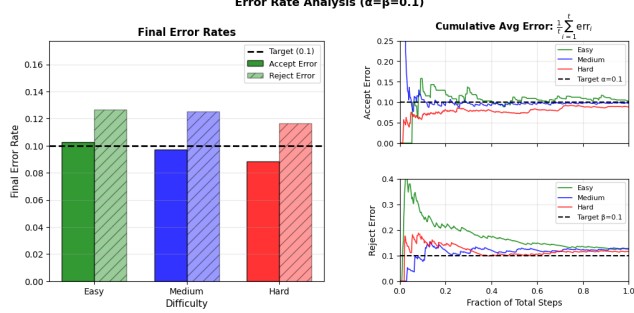

*Figure 5.* **Best-of-$n$ (MATH): error convergence for $\alpha = \beta = 0.10$.**

| Diff. | $\eta_A/\eta_R$ | $\tau_{A,0}/\tau_{R,0}$ | Err (A/R) |
|-------|-----------------|------------------------|-----------|
| Lvl 2 | 0.005 / 0.008   | 0.50 / 0.40            | .103 / .127 |
| Lvl 3 | 0.010 / 0.001   | 0.90 / 0.15            | .097 / .125 |
| Lvl 5 | 0.040 / 0.010   | 0.60 / 0.10            | .088 / .117 |

*Table 3.* Hyperparameters and final errors for Figure 5 ($\alpha = \beta = 0.10$).

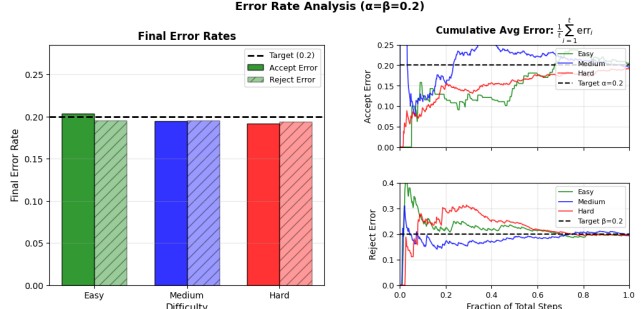

*Figure 6.* **Best-of-*n* (MATH): error convergence for $\alpha = \beta = 0.20$.**

| Diff. | $\eta_A/\eta_R$ | $\tau_{A,0}/\tau_{R,0}$ | Err (A/R) |
|---|---|---|---|
| Lvl 2 | 0.020 / 0.001 | 0.45 / 0.40 | .204 / .196 |
| Lvl 3 | 0.050 / 0.020 | 0.60 / 0.30 | .195 / .196 |
| Lvl 5 | 0.040 / 0.010 | 0.60 / 0.10 | .192 / .194 |

*Table 4.* Hyperparameters and final errors for Figure 6 ($\alpha = \beta = 0.20$).

### B.1.2. STEP-BY-STEP VERIFICATION (SUDOKU)

In the sequential Sudoku task, the algorithm must adapt to a non-stationary environment where the distribution of weak scores changes as the board reaches completion. Below, we provide convergence plots and parameter tables for $\alpha = \beta = 0.10$ and $\alpha = \beta = 0.05$. These results confirm that the "two-threshold" policy effectively intercepts errors step-by-step, maintaining a stable error rate over the stream of sequential verification decisions.

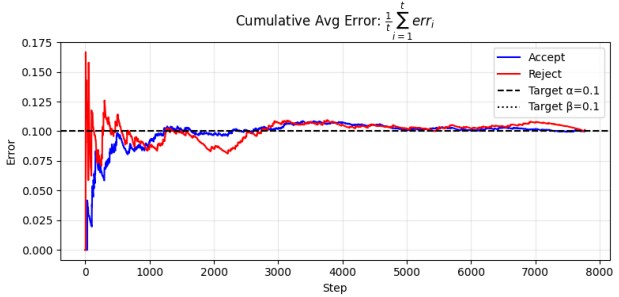

*Figure 7.* Error convergence for Sudoku ($\alpha = \beta = 0.10$).

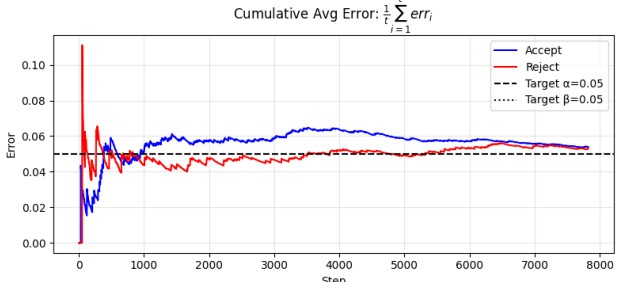

*Figure 8.* Error convergence for Sudoku ($\alpha = \beta = 0.05$).

| $\eta_A/\eta_R$ | $\tau_{A,0}/\tau_{R,0}$ | $\tau_{A,f}/\tau_{R,f}$ | Err (A/R) |
|---|---|---|---|
| 0.005 / 0.02 | 0.90 / 0.10 | 0.80 / 0.24 | .100 / .101 |

*Table 5.* Parameters for Sudoku ($\alpha = \beta = 0.10$).

| $\eta_A/\eta_R$ | $\tau_{A,0}/\tau_{R,0}$ | $\tau_{A,f}/\tau_{R,f}$ | Err (A/R) |
|---|---|---|---|
| 0.05 / 0.05 | 0.95 / 0.05 | 0.988 / 0.147 | .054 / .053 |

*Table 6.* Parameters for Sudoku ($\alpha = \beta = 0.05$).

### B.2. Detailed Reasoning-Cost Tradeoffs

We next refine the reasoning–cost analysis by decoupling the error targets $\alpha$ and $\beta$. In the main text, we swept targets under the constraint $\alpha = \beta$, which provides a convenient one-parameter summary of the accuracy–cost tradeoff. Here we perform *one-sided sweeps*: we fix one target and vary the other. This produces a *family* of tradeoff curves and lets us see how the operating frontier changes when a service provider prioritizes one type of reliability over the other.

**Why one-sided sweeps produce different curves.** At the population level, the Pareto formulation in Section 4 makes clear that type-I and type-II errors enter the objective with separate weights ($\lambda_1, \lambda_2$) (cf. (1)). Varying these weights changes the relative penalty assigned to accepting incorrect responses versus rejecting correct ones, and therefore changes where the optimal policy places its two thresholds. In our sequential algorithm, ($\alpha, \beta$) play an analogous role: smaller $\alpha$ forces

the accept threshold $\tau_A$ upward, shrinking the accept region to reduce incorrect acceptances; smaller $\beta$ forces the reject threshold $\tau_R$ downward, shrinking the reject region to reduce incorrect rejections. Fixing one target therefore fixes the corresponding constraint pressure, while sweeping the other moves the remaining threshold and traces a different tradeoff curve.

The key observation across these plots is the variation in slope and curvature as the fixed target changes. Figure 9 shows tradeoff curves obtained by holding $\alpha$ fixed and sweeping $\beta$, for three difficulty subsets. Each line corresponds to a different fixed $\alpha$, and points along a line correspond to different $\beta$ values. Figure 10 reports the complementary experiment, where $\beta$ is fixed and $\alpha$ is swept. Here, the fixed $\beta$ anchors the reject threshold $\tau_R$, while varying $\alpha$ primarily controls how much of the high-score region can be accepted without strong verification. Figure 11 shows the analogous one-sided sweeps for step-by-step Sudoku.

The additional plots in this appendix show that different fixed values of $\alpha$ or $\beta$ lead to different Pareto frontiers, enabling finer-grained operating points. Decoupling $(\alpha, \beta)$ reveals that there is not a single accuracy–cost frontier, but a family of frontiers indexed by which error type is held fixed. This gives a service provider a more precise control knob for selecting an operating point that matches task requirements and deployment constraints.

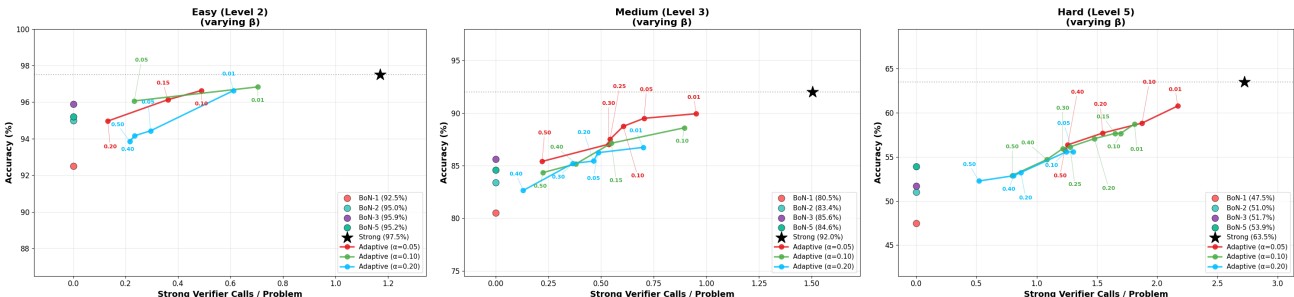

*Figure 9.* **Best-of-$n$ (MATH): one-sided sweeps with fixed $\alpha$ and varying $\beta$.** Reasoning accuracy vs. verification cost (strong-verification usage) for three difficulty subsets (left to right: Levels 2, 3, and 5). Each curve corresponds to a fixed value of $\alpha$ (type-I target), and points along a curve are obtained by sweeping $\beta$ (type-II target), tracing a family of accuracy–cost tradeoffs.

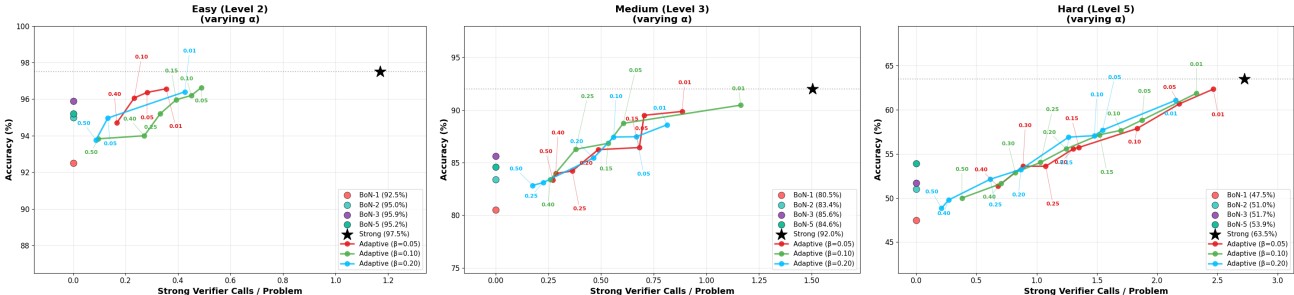

*Figure 10.* **Best-of-$n$ (MATH): one-sided sweeps with fixed $\beta$ and varying $\alpha$.** Reasoning accuracy vs. verification cost for three difficulty subsets (left to right: Levels 2, 3, and 5). Each curve corresponds to a fixed value of $\beta$ (type-II target), and points along a curve are obtained by sweeping $\alpha$ (type-I target). Together with Figure 9, these plots show how asymmetric error requirements induce different tradeoff frontiers.

### B.3. Analysis of Weak Verifier Scores: Sharpness, Calibration, and the Efficiency Frontier

To better interpret the accuracy–cost frontiers in Section 6, we analyze the weak verifier scores on the MATH outcome-level verification stream. The population analysis in Section 4 highlights two properties that govern whether weak scores can reduce strong-verification usage without sacrificing reliability: *sharpness* (scores are decisive) and *calibration* (scores are interpretable). This appendix section quantifies both and connects them to the observed tradeoff curves.

**Sharpness.** Sharpness refers to the ability of the weak verifier to produce decisive scores near 0 (confidently incorrect) or 1 (confidently correct), effectively separating the distributions of the scores. In this section, we quantify sharpness using the metric $|Score - 0.5|$, where 0 represents total uncertainty and 0.5 represents maximum confidence.

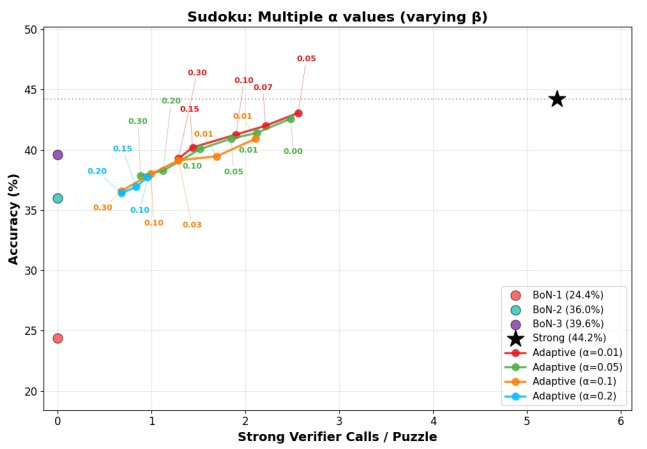

(a) **Fixed $\alpha$, sweep $\beta$.** Each line fixes the type-I target $\alpha$; points sweep $\beta$, yielding different accuracy–cost curves.

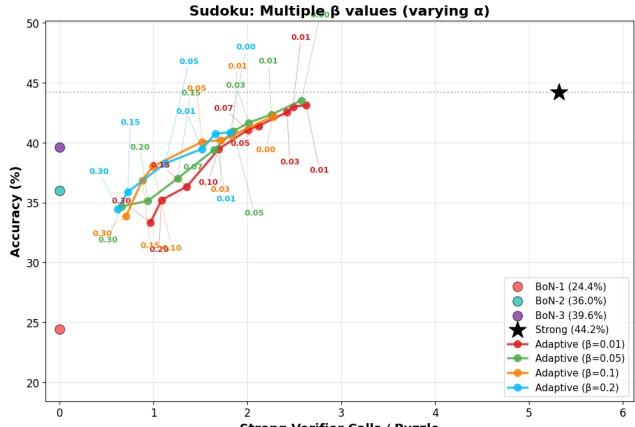

(b) **Fixed $\beta$, sweep $\alpha$.** Each line fixes the type-II target $\beta$; points sweep $\alpha$, yielding different accuracy–cost curves.

*Figure 11.* **Step-by-step (Sudoku): asymmetric accuracy–cost tradeoffs under one-sided sweeps.** Decoupling $(\alpha, \beta)$ produces a family of operating curves rather than a single frontier (cf. the main-text sweeps with $\alpha = \beta$).

As shown in Table 7 and the corresponding distributions in Figure 12, the weak verifier for the Easy and Medium datasets exhibits exceptionally high sharpness. The mean sharpness values ($0.467$ and $0.448$) and high separation scores ($0.57$ and $0.54$) indicate the verifier is "certain" about its assessment for most queries. This decisive behavior is the primary driver for the relatively steep slopes observed in the accuracy-cost tradeoff curves in Section 6. Because scores are concentrated at the extremes, adaptive thresholds can safely accept or reject a large portion of the query stream without manual intervention, achieving near-Oracle accuracy with significantly reduced strong verification calls.

In contrast, the Hard dataset demonstrates significantly lower sharpness (Mean = $0.358$). As visualized in the score distributions in Figure 13, the overlap between correct and incorrect responses is much larger in this regime, with a separation of only $0.37$. This lack of sharpness explains the more linear slope in the Level 5 tradeoff curves. When the weak signal is ambiguous, the algorithm identifies that the risk-weighted cost of an automated mistake is high, forcing the policy to defer to the strong verifier more frequently.

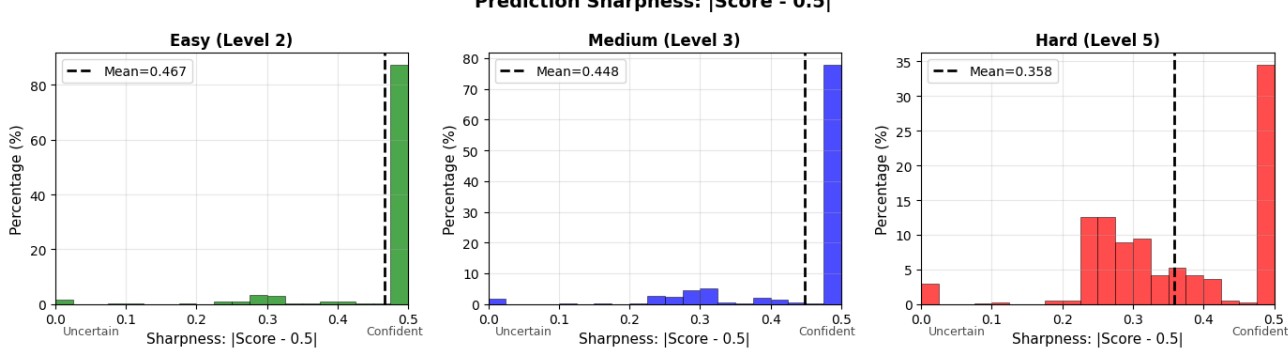

*Figure 12.* **Sharpness distributions across MATH difficulty.** Histograms of $|w - 0.5|$: higher mass near $0.5$ indicates more decisive weak scores.

*Table 7.* **Sharpness of weak verifier scores across MATH difficulty.** We report summary statistics of the sharpness metric $|w - 0.5|$ (range $[0, 0.5]$), where larger values indicate more decisive weak scores.

| Difficulty | Mean | Median | Std. |
|---|---|---|---|
| Easy (Level 2) | 0.467 | 0.496 | 0.084 |
| Medium (Level 3) | 0.448 | 0.495 | 0.100 |
| Hard (Level 5) | 0.358 | 0.342 | 0.120 |

Score Distribution: Correct vs Incorrect (Discrimination)

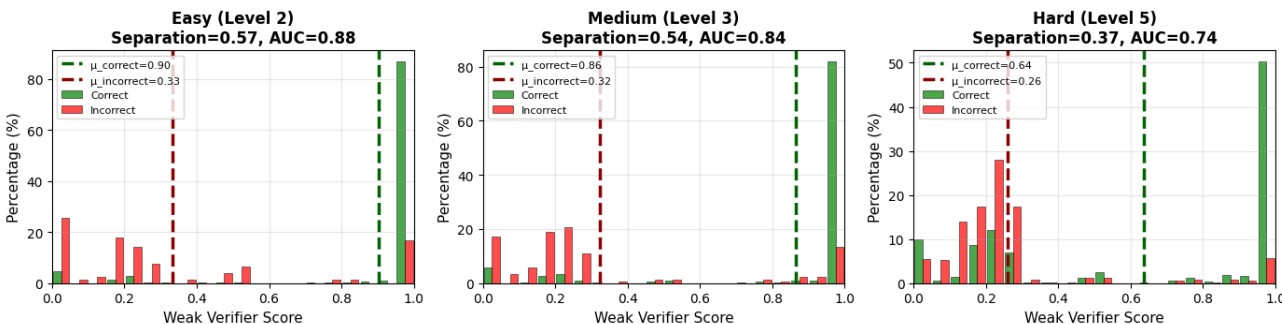

*Figure 13.* **Weak-score separation for correct vs. incorrect responses.** Score distributions conditioned on $g(P, R) = 1$ vs. $g(P, R) = 0$; greater overlap implies weaker discrimination and higher reliance on strong verification.

Calibration: Predicted Score vs Actual Accuracy

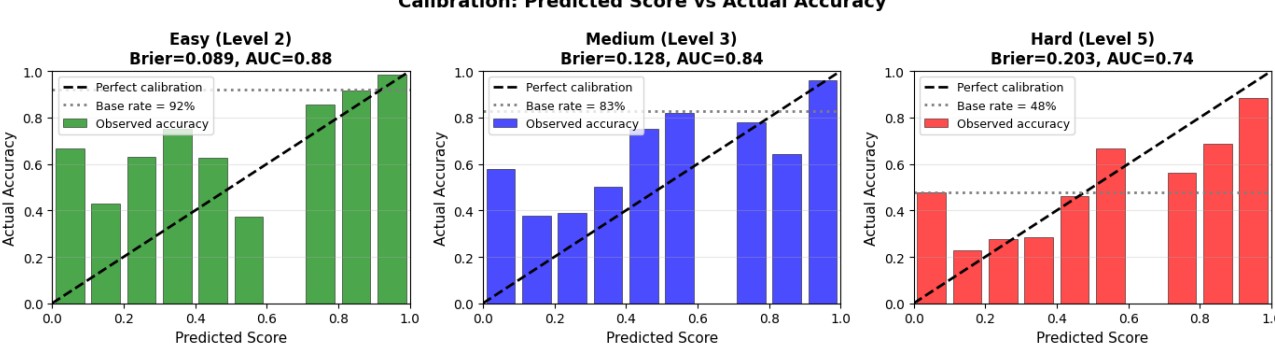

*Figure 14.* **Weak-score calibration on MATH.** Comparing empirical correctness rates to weak scores (perfect calibration lies on the diagonal).

**Calibration.** Beyond sharpness, weak scores must correlate with correctness. Table 8 reports standard discrimination and calibration statistics, including AUC and Brier score, along with the mean weak score for correct and incorrect responses and their gap ("separation"). Figure 13 visualizes the score distributions conditioned on $g(P, R) \in \{0, 1\}$.

The Easy and Medium subsets show stronger separation between correct and incorrect responses (Table 8) and clearer conditional distributions (Figure 13), which supports reliable accept/reject decisions at the extremes of the score range. For Hard, the conditional distributions overlap substantially and separation decreases, so the weak score is less informative and strong verification remains necessary more often. Finally, Figure 14 summarizes calibration by comparing predicted score ranges to empirical correctness rates.

*Table 8.* **Weak verifier discrimination and calibration diagnostics on MATH. AUC** measures discrimination between correct vs. incorrect responses (higher is better). **Brier** is mean squared error between $w$ and $g$ (lower is better). $\mu_{\text{correct}}$ and $\mu_{\text{incorrect}}$ are mean scores conditioned on $g = 1$ and $g = 0$, and **Separation** is $\mu_{\text{correct}} - \mu_{\text{incorrect}}$.

| Difficulty | Base Acc. | AUC | Brier | $\mu_{\text{correct}}$ | $\mu_{\text{incorrect}}$ | Separation |
|---|---|---|---|---|---|---|
| Easy (Level 2) | 92.2% | 0.88 | 0.089 | 0.90 | 0.33 | 0.57 |
| Medium (Level 3) | 82.7% | 0.84 | 0.128 | 0.86 | 0.32 | 0.54 |
| Hard (Level 5) | 47.9% | 0.74 | 0.203 | 0.64 | 0.26 | 0.37 |

### B.4. Performance across varying generators and weak verifiers

Our theoretical guarantees place *no* assumption on the quality of the generator or the weak verifier. In practice, however, the two components play distinct roles, and it is useful to separate them. The weak verifier governs the *reliability–cost tradeoff*: the more informative it is, the larger the fraction of the stream on which SSV can accept or reject without consulting the strong verifier, and hence the more strong-verification cost is saved. If the weak verifier carries little signal, there is no

alternative but to rely more heavily on the strong verifier; the role of SSV is therefore not to make a weak verifier strong, but to exploit it whenever it is sufficiently informative. The generator plays a different role: it affects whether the underlying task can be solved at all, rather than the verification tradeoff. With a poor generator, most candidates are incorrect and will be rejected regardless of which verifier is used. The two experiments below isolate these effects of generators and weak verifiers of varying quality, while in all cases SSV continues to control the target error rates $\alpha, \beta$.

**varying underlying generator.** We repeat the MATH evaluation while varying the solution generator, holding the weak–strong verifier pair fixed. As argued above, generator capability should move the achievable accuracy of the task without compromising the reliability guarantee. Figure 15 confirms this: across generators of differing capability, the running type-I and type-II errors stabilize at the nominal targets, while the accuracy–cost frontier shifts with generator quality. Stronger generators produce a larger fraction of correct candidates and therefore reach higher accuracy at the same strong-verification budget, but the *shape* of the tradeoff (how much cost SSV saves relative to the Strong-Only oracle) is determined by the weak verifier rather than the generator.

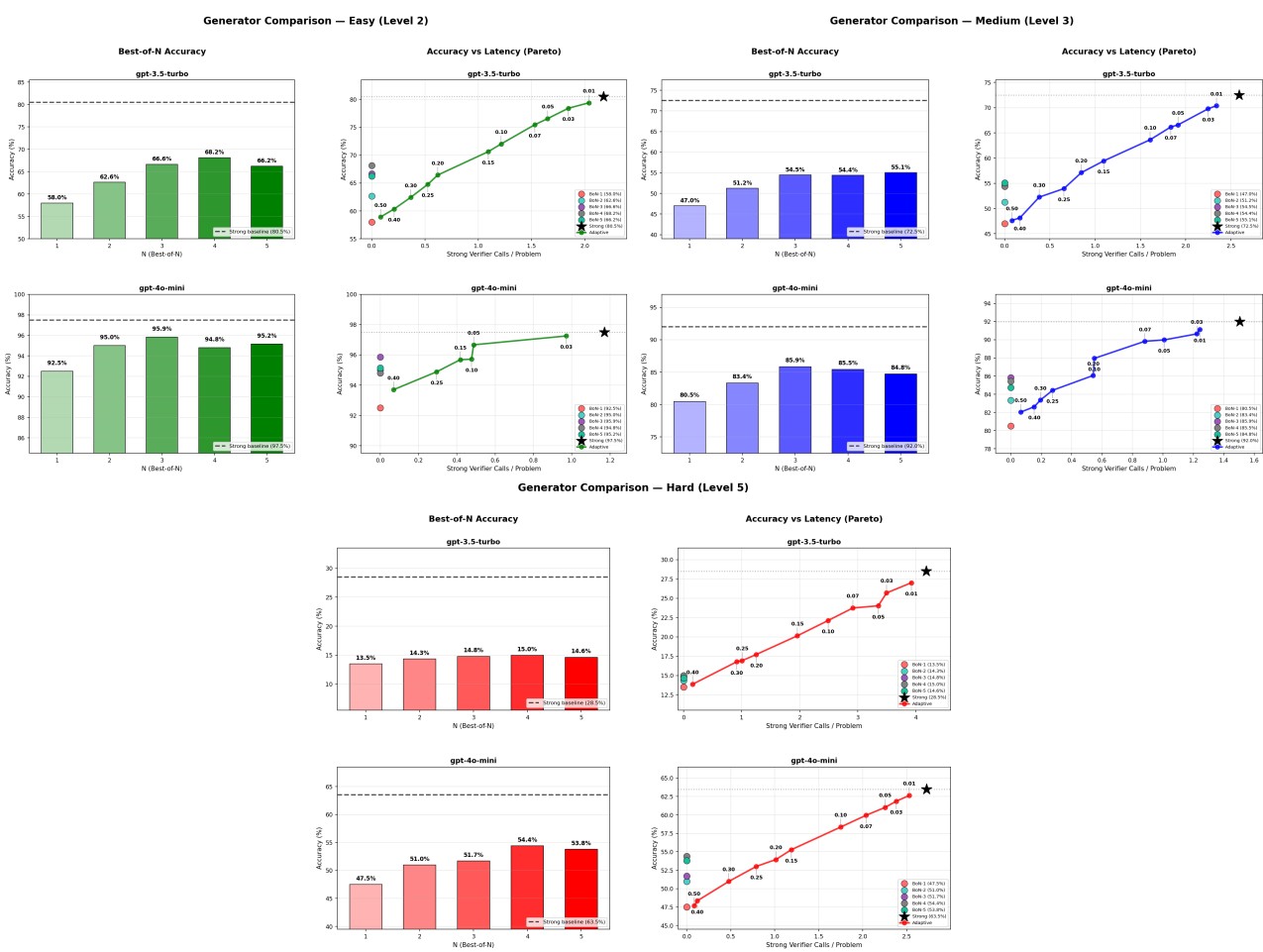

*Figure 15.* Generator comparison across difficulty levels. GPT-4o-mini (bottom row) achieves substantially higher Best-of-N accuracy than GPT-3.5-turbo (top row) at every N, reflecting its stronger generation quality. This translates to a more favorable accuracy–latency tradeoff in the Pareto curves. Notably, the adaptive algorithm recovers most of the strong baseline accuracy for both generators, but does so at lower cost when the underlying generator is stronger.

**varying underlying weak verifier.** We next fix the generator and strong verifier and vary the weak verifier across instances of differing quality. For each weak verifier we report its calibration and sharpness, using the same diagnostics as Appendix B.3, alongside the realized strong-verification cost and accuracy of SSV. Figure 16 show that more informative weak verifiers, i.e. those with higher sharpness and better calibration, allow SSV to avoid strong verification on a larger fraction of the stream, producing steeper accuracy–cost frontiers and larger savings, whereas weak verifiers with little

instance-specific signal force heavier reliance on the strong verifier and flatter frontiers. In every case the target error rates are maintained; weak-verifier quality affects *how much* cost can be saved, not *whether* reliability holds.

### B.5. Robustness to Distribution Shift and Non-Stationary Streams

The guarantees in 5 is distribution-free and holds for *arbitrary* query streams. We conduct two experiments that stress-test that our guarantees hold under arbitrary distribution shifts. First, to test robustness to weak-verifier drift, we inject a weak-verifier distribution shift mid-stream: after a fixed number of rounds the weak verifier is replaced by another weak verifier of varying quality, simulating changing proxy signal or replacement of the proxy signal. As shown in Figure 17, SSV detects the change and re-adapts its thresholds $(\tau_R^t, \tau_A^t)$, with the running type-I and type-II errors re-converging to the targets under the new regime after a short transient; because the threshold updates are driven only by observed strong labels, no knowledge of the shift time or of the new verifier is required. Second, to test robustness to non-i.i.d. data streams, we construct non-stationary MATH streams by concatenating the three difficulty subsets (Easy/Level 2, Medium/Level 3, Hard/Level 5) and sweeping over all orderings of the three difficulty blocks, inducing abrupt, structured shifts in the distribution of weak scores as the stream transitions between difficulty regimes. Figure 18 shows that the running type-I and type-II errors converge to the targets under every ordering, including the most adversarial cases where the hardest block appears first: the transient adaptation behavior depends on the ordering, but the long-run error control does not, consistent with Theorem 5.1.

### B.6. Verifier Implementation Details and Prompt Specifications

Across both tasks, we employ a multi-agent LLM pipeline to generate, score, and verify solutions. For the outcome-level (MATH) task, We utilize GPT-4o-mini (OpenAI, 2024) as the solution generator and DeepSeek-Chat (DeepSeek-AI, 2024) as the weak verifier providing a continuous confidence signal $w \in [0, 1]$ used by our adaptive policy. The strong verifier for the MATH dataset is GPT-4o, which assesses mathematical equivalence between generated answers and ground-truth labels.

For the process-level (Sudoku) task, we employ GPT-4o-mini as the step-by-step move generator and DeepSeek-Chat as the weak verifier. Unlike the math task, the strong verifier for Sudoku is a deterministic programmatic check that validates moves directly against the unique ground-truth solution of the $4 \times 4$ grid, removing the need for an additional LLM for ground-truth verification.

Below we provide all prompt templates for both tasks; each is titled and categorized for clarity.

---

**Generator (GPT-4o-mini) — Prompt for Full-Solution Generation (MATH)**

```
Solve this math problem step by step.  Show your reasoning, then give the final
answer.

PROBLEM: {{problem}}

Return JSON only:
{
  "reasoning":  "<step-by-step reasoning>",
  "final_answer":  "<final answer only (e.g., 42, 3/4, x=5)>"
}

CRITICAL RULES:
- Return only the JSON object (no extra text).
- final_answer must be just the final answer (no explanation).
```

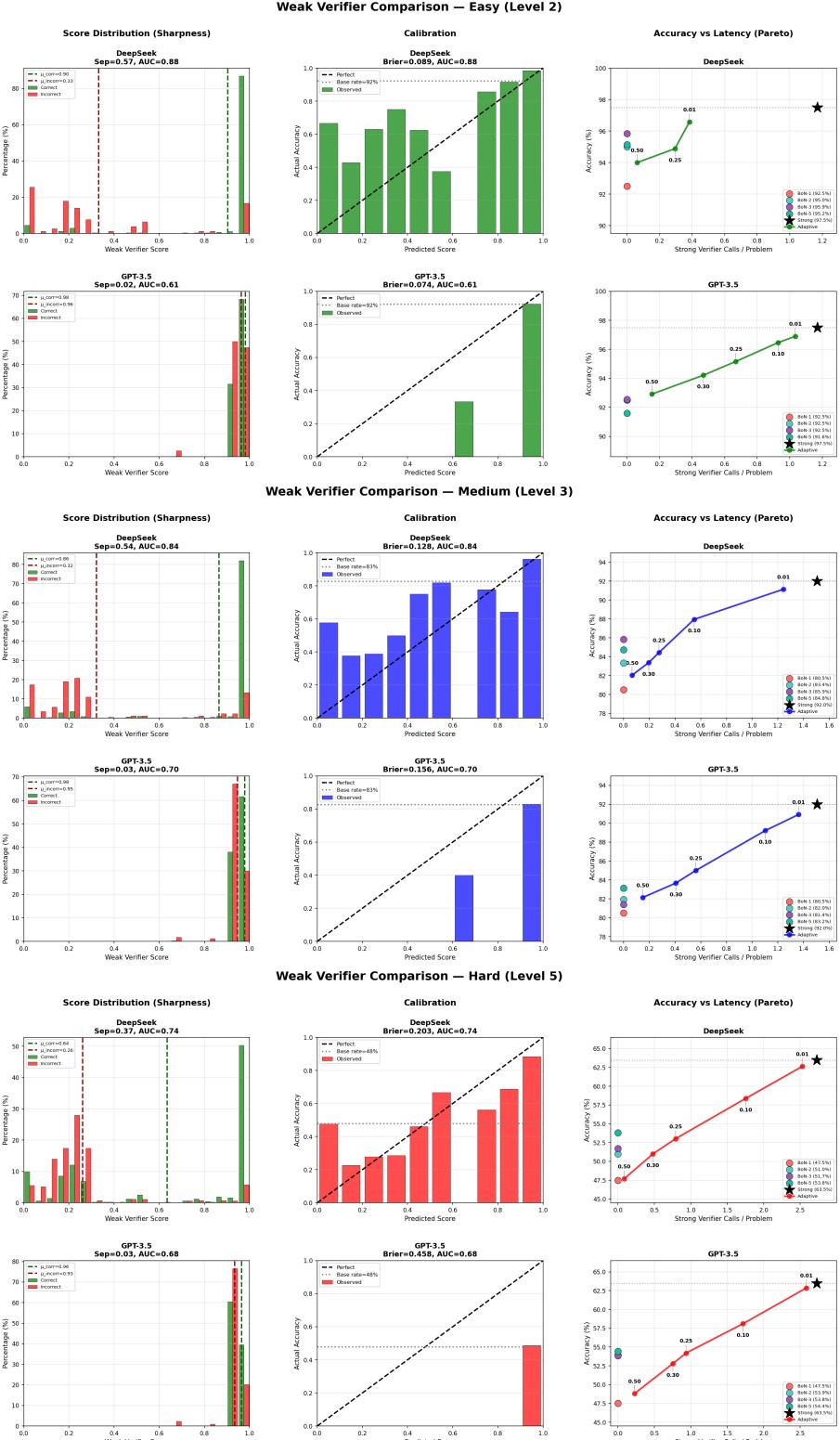

*Figure 16.* Comparison of two weak verifiers (DeepSeek vs GPT-3.5-Turbo) scoring the same generated solutions. The left column shows score sharpness — how well the verifier separates correct from incorrect solutions. The center column shows calibration: how closely predicted confidence aligns with actual accuracy. A verifier with higher sharpness and better calibration produces a more informative signal, which the adaptive algorithm can exploit to make better accept/reject decisions early. This manifests directly in the Pareto curves (right column): the sharper, better-calibrated verifier (DeepSeek here) achieves the same accuracy with fewer strong verifier calls, while the weaker verifier forces the algorithm to fall back on expensive strong verification more often to maintain the same guarantees.

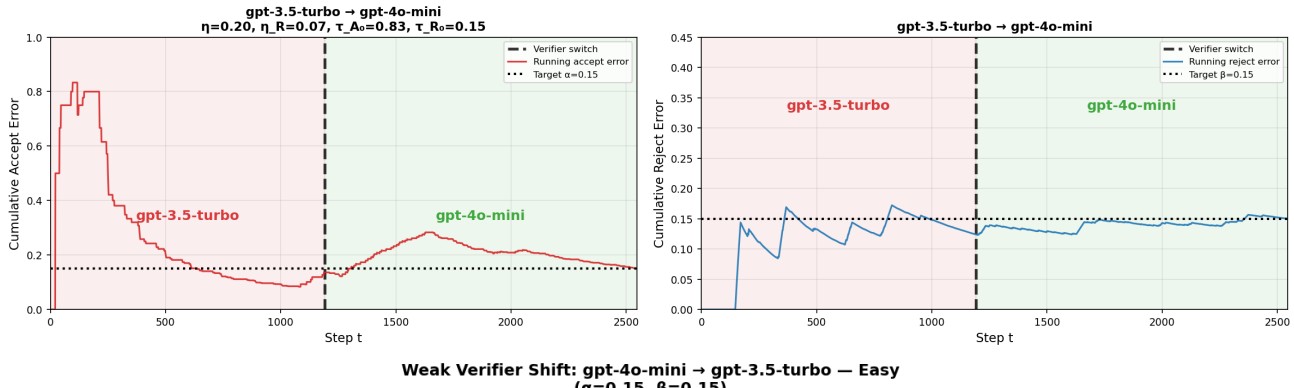

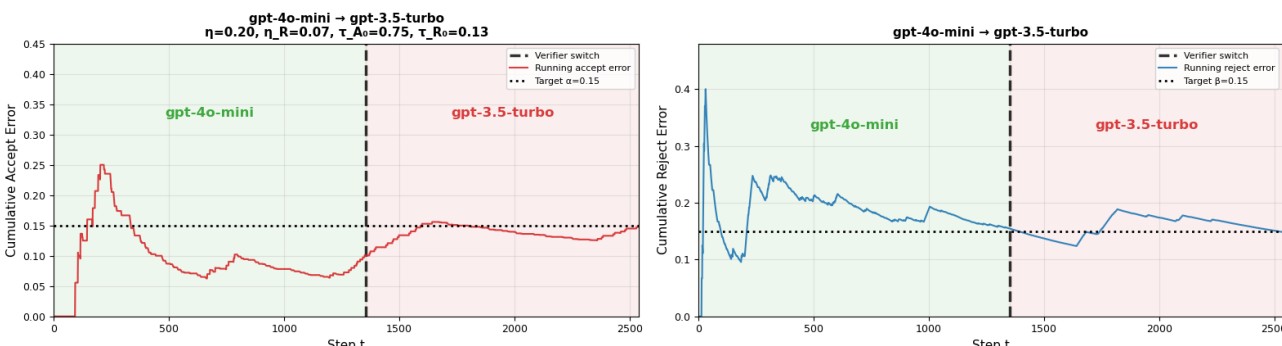

*Figure 17.* We test our algorithm under abrupt shift of the weak verifier signal ( where the underlying weak verifier is changed half way through the stream, causing abrupt distribution shift. ) Our framework is able to satisfy the desired error rates. We consider both ordering of the weak verifier signals in the stream for robustness analysis.

*Figure 18.* Running type-I and type-II error convergence under different orderings of the Easy/Medium/Hard difficulty blocks. SSV converges to the target error rates under all stream orderings

---

**Weak Verifier (DeepSeek-Chat) — Prompt for Scoring a Solution (MATH)**

```
Rate the correctness of this math solution on a scale from 0.00 to 1.00.

PROBLEM: {{problem}}

REASONING:
{{reasoning}}

FINAL ANSWER: {{final_answer}}

Scoring guidance:
1.  Is the reasoning mathematically sound?
2.  Are all calculations correct?
3.  Does the final answer follow from the reasoning?

Return JSON only (nothing else):
{
  "score":  0.XX
}
```

---

**Strong Verifier (GPT-4o) — Prompt for Answer Equivalence Checking (MATH)**

```
Determine whether two math answers are equivalent.

PROBLEM: {{problem}}

STUDENT'S REASONING:
{{reasoning}}

STUDENT'S ANSWER: {{generated_answer}}
CORRECT ANSWER: {{ground_truth_answer}}

Equivalence guidance:
- Different formats can be equivalent (e.g., 5, 5.0, 5/1, \boxed{5}).
- Simplified vs. unsimplified fractions can be equivalent (e.g., 2/4 = 1/2).
- Notation differences can be equivalent when implied by the question (e.g., x=3 vs.
3).

Return JSON only:
{
  "is_correct":  true or false,
  "explanation":  "<brief reason>"
}
```

**Generator (GPT-4o-mini) — System Prompt (Sudoku: one-move-at-a-time)**

```
You solve 4x4 mini Sudoku by proposing ONE move at a time.

Rules:
- Grid is 4x4 with 2x2 boxes.
- Fill each cell with 1--4.
- Each row/column/2x2 box must contain 1,2,3,4 exactly once.
- 0 means empty.

Return JSON only with keys:  row, col, value, why, confidence.
{
  "row":   0--3,
  "col":   0--3,
  "value":  1--4,
  "why":   "One short LOCAL deduction (row/col/box).",
  "confidence":  0.00--1.00
}

CRITICAL: Output only JSON. No extra keys.  No markdown.
```

**Weak Verifier (DeepSeek-Chat) — Prompt for Scoring a Cell Placement (Sudoku)**

```
You are scoring a proposed move in 4x4 Sudoku.

Current state:  {{current_grid}}
Previous moves:  {{previous_steps}}

Proposed move:  place {{value}} at ({{row}}, {{col}}).

Question:  What is the probability (0.00 to 1.00) that this value is the correct
value for this cell?
Consider legality, whether logic forces this value, and whether alternatives remain.

Return only a single decimal number in [0.00, 1.00].
```

