# OpenReview forum: "When to Trust the Cheap Check: Weak and Strong Verification for Reasoning"
_ICML.cc/2026/Conference — ICML 2026 spotlight_

### Official Review · Reviewer_Acq5 · 2026-02-27

**Soundness:** 3
**Presentation:** 3
**Significance:** 3
**Originality:** 3
**Overall Recommendation:** 5
**Confidence:** 3

**Summary:**

This paper studies how to orchestrate two kinds of verification that commonly appear in LLM reasoning systems: (1) cheap, noisy “weak verification” signals such as self-consistency, proxy rewards, or learned critiques, and (2) costly but reliable “strong verification” such as human inspection or domain-grounded execution. The authors formalize a weak–strong verification policy that decides, for each prompt–response pair, whether to accept, reject, or defer to strong verification based only on the weak score. They propose metrics capturing incorrect acceptance (type-I), incorrect rejection (type-II), and the frequency of strong verification calls. Under population-level assumptions they show optimal policies have a two-threshold structure and analyze when weak verifiers are valuable in terms of calibration and sharpness. Building on these insights, they introduce an online algorithm (SSV) that aims to control type-I/type-II errors in a fully distribution-free manner, with no assumptions about the query stream, the LLM, or the weak verifier. Experiments on mathematical reasoning and a process-level sequential decision-making task suggest the algorithm can approach the reliability of always using strong verification while substantially reducing the number of strong-verification calls.

**Compliance With Llm Reviewing Policy:**

Affirmed.

**Final Justification:**

This paper addresses an important and timely problem in LLM reasoning systems: how to combine cheap but noisy weak verification signals with costly but reliable strong verification in a principled way. The formulation is clear and useful. By defining a weak–strong verification policy with three actions: accept, reject, or defer to strong verification, the paper captures a practically meaningful reliability–cost trade-off. The population-level two-threshold characterization is intuitive and helps explain what an optimal policy should look like, while the proposed online SSV algorithm adds a stronger contribution by aiming to control incorrect acceptance and rejection in a distribution-free sequential setting.

The paper is technically solid overall. Its main strength is that it moves beyond heuristic verification orchestration and offers a more rigorous decision framework for when weak signals can be trusted. The distinction between type-I error, type-II error, and strong-verification frequency is well chosen, and the online guarantees, if they hold as stated, represent a meaningful step forward for reliable LLM system design. The empirical results are also encouraging, suggesting that the method can approach the reliability of always invoking strong verification while substantially reducing the number of expensive strong-verification calls.

The main weaknesses are not fundamental, but they do limit how broadly the results can currently be interpreted. In practice, strong verification is often imperfect, noisy, delayed, or incomplete, whereas the paper’s framework appears to treat it as a reliable reference point. Similarly, while the method is advertised as distribution-free and sequential, the experimental evidence would be stronger with more explicit stress tests under nonstationarity, adversarial streams, or weak-verifier drift. The paper could also do more to clarify practical deployment details, such as hyperparameter selection, online statistics to maintain, computational overhead, and small-sample behavior.

Overall, I support acceptance. The paper makes a meaningful conceptual and technical contribution by formalizing the weak-versus-strong verification decision layer and providing a principled online method for controlling verification errors under minimal assumptions. Its framing is broadly relevant to LLM reasoning pipelines, and the results suggest real practical value for reducing verification cost without giving up much reliability. While additional discussion of noisy strong verifiers and robustness under realistic deployment conditions would strengthen the work further, the paper is already strong enough to merit acceptance.

**Key Questions For Authors:**

1. How does the method behave when the strong verifier is imperfect or noisy (e.g., human disagreement, incomplete execution checks)? Can your guarantees or empirical behavior be extended to “strong but noisy” verification?

2. Your setting is explicitly distribution-free and allows arbitrary query streams, potentially dependent on past decisions. Can you include experiments that explicitly construct adversarial or heavily shifting streams to validate robustness beyond i.i.d.-like benchmarks?

3. How sensitive is performance to miscalibration of the weak verifier over time (e.g., weak verifier drift due to prompt changes or model updates)? Do you observe degradation, and does the algorithm adapt fast enough in such nonstationary settings?

4. Can you provide clearer operational guidance on hyperparameters and overhead: what is the per-step computation, what statistics must be tracked, and what is the sample complexity needed before the thresholds become reliable?

5. The paper motivates calibration and sharpness as governing the value of weak verifiers. Can you provide additional empirical evidence connecting measured calibration/sharpness to realized cost savings across different weak verifiers and tasks?

**Limitations:**

Partially. The paper frames the setting clearly and provides strong claims about distribution-free online control, but it would benefit from a more explicit discussion of practical limitations: strong verification may be noisy/partial/delayed; weak verifiers may drift; and type-I/type-II control with respect to a verifier does not always translate to real-world utility if the verifier does not fully capture correctness. Additionally, the method’s benefit may be reduced when strong verification is required frequently due to inherently low-quality weak signals, and this regime should be highlighted as a boundary case.

**Strengths And Weaknesses:**

Soundness: The problem formulation is timely and well motivated: many systems already combine weak internal signals with occasional external verification, but lack principled guarantees about when weak checks should be trusted. The paper’s formalization is clean (three actions: accept, reject, defer) and the three metrics capture a meaningful reliability–cost trade-off. The population analysis providing a two-threshold structure is intuitive and helps clarify what one should expect from an optimal policy. The most compelling technical contribution is the distribution-free online control of acceptance and rejection errors; if the guarantees are correct as stated, this is a strong step beyond purely heuristic calibration.
That said, the practical soundness depends heavily on how the “strong verifier” is instantiated and whether its judgments are stable and unambiguous. In many real deployments, strong verification can be expensive and also imperfect (humans disagree; execution checks can be incomplete; environment feedback may be delayed). The paper would benefit from a clearer discussion of what happens when the strong signal is noisy or only partially observed. Additionally, it is important to ensure that the online guarantees correspond to the exact operational objective; for example, controlling type-I/type-II errors relative to a strong verifier is not necessarily identical to controlling end-to-end task success when downstream decisions interact with the accept/reject outcomes. Finally, while the experiments are promising, I would like to see stronger stress tests under distribution shift and adversarial streams, since the stated setting is distribution-free and sequential feedback can create challenging dependence.

Presentation: The paper is well written and the core concepts (weak vs strong verification, policy actions, and metrics) are easy to follow. The separation between population analysis (explaining threshold structure and the role of calibration/sharpness) and the online algorithm is a good narrative. Where the paper could improve is in providing more “engineering clarity” for readers who want to apply the algorithm: what statistics must be tracked online, how hyperparameters are chosen, what the computational overhead is, and how the method behaves in small-sample regimes. A more explicit explanation of the experimental protocol for strong verification costs (e.g., how strong verification budget is modeled, what a “call” entails, and how latency/cost is measured) would help make the cost frontier claims more concrete.

Significance: The contribution is potentially significant because verification orchestration is a core bottleneck in making LLM reasoning reliable at scale. A method that can provably control incorrect acceptance and rejection while reducing reliance on expensive verification could influence both research and practical system design. The paper’s framing also unifies multiple lines of work (selective prediction, learning-to-defer, calibration) in a way that is directly relevant to LLM pipelines. The main limitation is that real-world strong verification is often messy (noisy, delayed, partial), and weak verifiers can drift as the LLM or prompting changes; the significance will be strongest if the paper convincingly demonstrates robustness to these realistic complications.

Originality: The paper’s novelty is less about introducing a new weak verifier and more about formalizing and controlling the decision layer that decides when to trust weak checks and when to escalate to strong verification. The combination of (i) an explicit weak–strong policy framework, (ii) a two-threshold characterization under population assumptions, and (iii) a distribution-free online method that targets separate type-I/type-II control is a meaningful and fairly original synthesis. Even if some ingredients relate to prior selective prediction/L2D work, the adaptation and emphasis on weak-versus-strong verification in LLM reasoning is a valuable perspective.

---

> ### Author Rebuttal · Authors · 2026-03-30
>
> We thank the reviewer for the thoughtful and positive assessment. We are glad the reviewer found the problem timely, the formulation clean, the population analysis intuitive, and the distribution-free online guarantees compelling. We also appreciate the constructive suggestions and will incorporate these clarifications and additional experiments in the revision.
>
> **Q1. Strong verifier noise.**
>
> The strong verifier is meant to represent the highest standard of verification available, such as expert human inspection. At the same time, even such a verifier need not be literally infallible. Accordingly, our type-I and type-II errors are defined with respect to the strong verifier, not an absolute ground-truth oracle. If the strongest available verifier can still be wrong, then in many settings there is no better notion of correctness available at decision time, though delayed downstream feedback may sometimes help. We will clarify this point more explicitly in the formulation and discussion/limitations section. An earlier draft contained a longer discussion of this, but we had to shorten it for space.
>
> **Q2. On stress testing and non-iid streams**
>
> We agree this would be valuable. Since our theory is explicitly distribution-free and allows arbitrary streams, such stress tests are fully aligned with the paper. In the camera ready version, we will add experiments with several different shifted streams. For now, we have added the following results to the revised version.
>
> We vary the difficulty ordering of the MATH problems across all possible orderings of the three difficulty levels and show that our method converges to the target error rates under all stream orderings **Results: https://imgur.com/a/hssp5rx.**
>
> **Q3. Weak-verifier drift.**
>
> Theoretically, our guarantees do not require any assumption on the quality or stationarity of the weak verifier. Practically, however, weak-verifier quality directly affects how much strong-verification cost can be saved: if the weak verifier becomes less informative, stronger reliance on the strong verifier is unavoidable. SSV is designed to adapt to such changes online, and we will clarify this in the revision. We are also adding experiments with additional weak verifiers.
>
> **To directly address robustness to weak verifier drift, we conducted additional experiment where we introduce a weak verifier distribution shift mid-stream** and show that our algorithm detects and adapts to the shift, converging to the target error rates under the new verifier regime. Results: https://imgur.com/a/OK5SOUY. We will add these to the revised version.
>
> **Q4. Hyperparameters and overhead.**
>
> We agree this deserves a clearer explanation. The policy itself is lightweight: it tracks only two thresholds and updates them only when strong verification is queried. The main hyperparameters are the step size and exploration probabilities, which govern the adaptation/stability and accuracy/cost tradeoffs. We will make these implementation details more explicit in the revision.
>
> **Q5. Calibration/sharpness and savings.**
>
> We agree this connection should be emphasized more clearly. Conceptually, this is already one of the main messages of the paper: calibration makes weak scores interpretable, while sharpness determines whether strong-verification usage can be reduced. Importantly, in this work we are not directly trying to design the best possible weak verifier. Rather, our goal is to understand what properties of a weak verifier make it useful, and to develop an algorithm that can make the best possible use of a given weak–strong pair.
>
>
>
> Please note that in the current manuscript, we provide a detailed analysis of these properties in Appendix B.3, where we quantify the sharpness and calibration of the weak verifier across MATH difficulty levels and directly connect them to the observed accuracy–cost tradeoffs.
>
> **That being said, to address your question directly we have conducted additional experiments comparing weak verifiers of varying quality**, with calibration and sharpness reported alongside realized cost savings, confirming that these properties are the primary drivers of achievable tradeoffs. Results: https://imgur.com/a/B4dXxSl. These results will be included in the revision.
>
> Finally, due to space constraints, we did not include a separate dedicated limitations subsection, though some limitations are already discussed. In the revision, we will make the limitations discussion more explicit and expand it to include the comments raised by the reviewer.

---

> > ### Author Rebuttal · Reviewer_Acq5 · 2026-04-03
> >
> > Thank you for the thoughtful rebuttal. The response addresses my main concerns well. I especially appreciate the clarifications regarding the role of the strong verifier, the added experiments on shifted streams and weak-verifier drift, and the stronger empirical connection between calibration/sharpness and achievable cost savings. The discussion of implementation details and limitations is also helpful. Overall, the rebuttal reinforces my original positive assessment of the paper and adequately addresses my main concerns.

---

### Official Review · Reviewer_wAFk · 2026-03-10

**Soundness:** 3
**Presentation:** 4
**Significance:** 4
**Originality:** 4
**Overall Recommendation:** 5
**Confidence:** 4

**Summary:**

To check whether LLMS is right, weak, and strong verification is used. Weak verification is fast and cheap but noisy, while strong verification is slow, expensive, and more reliable. So the goal is to know when to use strong verification versus weak verification. So they propose SSV, which allows weak verification to handle most cases on its own, while strong verification is called in 2 situations: one where the weak score falls in the middle zone, and the other where SSV randomly probes the confident regions to ensure they are actually correct.  If they are wrong, it tightens standards (shifting the threshold up) and escalates more.

**Compliance With Llm Reviewing Policy:**

Affirmed.

**Final Justification:**

I am satisfied with the authors response. my scores will remain unchanged

**Key Questions For Authors:**

1. The paper motivates strong verification as human inspection; expensive, reliable, and drawing on contextual knowledge beyond what text can capture. Why was GPT-4o chosen as the strong verifier instead, and how do the theoretical guarantees hold when the strong verifier is itself a probabilistic model that can be wrong, particularly on hard problems?

**Limitations:**

No, the assumption that the strong verifier is a reliable oracle, and the implications when it isn't, which we've already established, is a real issue given GPT-4o is used in experiments.
Also, the paper should explicitly warn that the guarantees are conditional on strong verifier reliability, not unconditional. If the strong verifier is itself a noisy model like GPT-4o, real-world error rates could significantly exceed the guaranteed bounds, with serious consequences in high-stakes settings.

**Strengths And Weaknesses:**

Soundness
The theoretical results are clean and well-constructed. Theorem 5.1 gives finite-time, distribution-free error control. The use of importance weighting to correct for randomized probing is technically justified, and the martingale concentration argument (Freedman's inequality) is appropriate in this setting. The assumption of calibration in Section 4 is clearly flagged as a population-level tool for deriving structure, not a requirement for the actual algorithm
However, the most significant experimental gap is the choice of GPT-4o as the strong verifier. The paper's own motivation frames strong verification as human inspection: expensive, reliable, and drawing on contextual knowledge beyond what can be captured textually. Yet in practice, GPT-4o is used as a stand-in, which is neither as reliable nor as grounded as an actual domain expert. This matters theoretically too: Theorem 5.1's guarantees assume the strong verifier is a reliable oracle, but GPT-4o is a probabilistic model that can be wrong. Errors in the strong verifier propagate into the threshold updates, yet the paper includes no sensitivity analysis on strong verifier quality and doesn't acknowledge this in the limitations section.

Presentation
The paper is well structured and easy to follow. The progression from population-level analysis to algorithm design to guarantees to experiments is logical and builds intuition naturally before introducing formalism. The two-threshold structure is explained clearly before Theorem 4.2, which helps the result land naturally. Figure 1 effectively illustrates the three-action decision architecture. Algorithm 1 is sufficiently detailed, and the appendix provides adequate supplementary material for readers who want to go deeper on the proofs and experimental details.

Significance
The paper is very significant. The problem is real and practically important; any production LLM deployment faces exactly this tradeoff between verification cost and reliability. The framework is general enough to apply to both outcome-level and process-level verification, covering a wide range of reasoning paradigms. The fact that SSV handles both settings with the same algorithm, simply by redefining what a query and response are, demonstrates genuine generality rather than narrow applicability

Originality
The individual components exist in prior literature, and the authors acknowledge this. The originality is in the combination: applying distribution-free online calibration to weak-strong verification with separate Type-I and Type-II control under partial feedback. The partial feedback aspect is the genuinely novel wrinkle; you only observe the strong signal when you choose to query it, making threshold adaptation meaningfully harder than standard conformal prediction settings. Beyond the algorithm, explicitly defining weak-strong verification as a distinct layer above reasoning and verifier design carves out a new sub-problem that previously had no principled treatment, which could influence how future work approaches verification in LLM systems.

---

> ### Author Rebuttal · Authors · 2026-03-28
>
> We thank the reviewer for the very positive and thoughtful assessment. We are glad the reviewer found the problem practically important, the framework general, the algorithmic contribution meaningful, and the presentation clear and well structured. We also thank the reviewer for raising the important point about the role and reliability of the strong verifier.
>
> **On the role of the strong verifier.**
>
> The strong verifier is intended to represent the highest standard of verification available to us. In practice, this could indeed be human inspection or another highly reliable but expensive verification procedure. At the same time, even such a verifier may not be literally infallible. Our type-I and type-II errors are defined with respect to the strong verifier, rather than with respect to an absolute ground truth oracle. This is an important modeling point: if the strong verifier is the highest standard of verification available, then in many realistic settings it is also the operational notion of correctness available to the system. If even that verifier can sometimes be wrong, then one must ask what source of truth is actually available. In some applications, downstream outcomes or delayed feedback may partially reveal this, but that lies beyond the present scope. We will add a more explicit discussion of this matter both in the problem formulation and in the discussion/limitations section in the revision. We note that an earlier draft of the paper contained a discussion along these lines, but we unfortunately had to shorten it due to space constraints.
>
> **On the experimental strong verifier.**
>
> We would also like to clarify a point about the experimental setup. In our experiments, GPT-4o is not used as the strong verifier in the sense of replacing ground-truth verification with a generic model judgment. Rather, the strong verification procedure is designed using *access to the ground truth*. For Sudoku, strong verification is implemented exactly by deterministic code. For MATH, the difficulty is that the final answer may be expressed in mathematically equivalent but syntactically different ways, so GPT-4o is only used to determine equivalence between the model’s final answer and the known ground truth answer (details are provided in Appendix B4).  The rational is that these procedures serve as a reliable proxy for expert human inspecting the answers. In other words, the use of GPT-4o here is not to provide an open-ended probabilistic judgment, but to operationalize strong verification with the help of the ground truth, as we know the ground truth in the experiment environment. Because of this, it is much closer in spirit to expert inspection than to using a generic LLM as a noisy verifier. This engineering choice is made to obtain the same outcome as if a human were hired as the strong verifier, without actually hiring anyone. We will clarify this distinction much more explicitly in the revision.
>
> **On guarantees when the strong verifier is imperfect.**
>
> The reviewer is absolutely right that the guarantees of Theorem 5.1 are conditional on the strong verifier. We will make this explicit in the revised paper. More precisely, the theorem controls the empirical type-I and type-II errors relative to the strong verification signal. Thus, if the strong verifier itself is systematically noisy, then the resulting guarantees are necessarily with respect to that signal, not with respect to some inaccessible absolute truth. We will emphasize this point clearly in the limitations section, especially for higher-stakes applications where verifier imperfections matter.

---

> > ### Author Rebuttal · Reviewer_wAFk · 2026-04-03
> >
> > My concerns have been adequately addressed.

---

### Official Review · Reviewer_xjLV · 2026-03-11

**Soundness:** 3
**Presentation:** 3
**Significance:** 2
**Originality:** 2
**Overall Recommendation:** 4
**Confidence:** 3

**Summary:**

This paper studies how to decide when a cheap but noisy weak verifier is sufficient and when a more reliable but costly strong verifier should be used during LLM reasoning. It proposes an online algorithm, Selective Strong Verification (SSV), that adaptively sets thresholds to control acceptance and rejection errors while reducing the number of strong-verifier calls. Empirically, the method achieves reliability close to exhaustive strong verification on math reasoning and Sudoku while using substantially less verification cost.

**Compliance With Llm Reviewing Policy:**

Affirmed.

**Final Justification:**

The authors address my concern during the rebuttal.

**Key Questions For Authors:**

* How is the learned threshold expected to generalize in practice? If the threshold is learned from one set of questions, is it still applicable to a new set of questions drawn from a different distribution or domain?
* What is the practical cost of updating the threshold online? Since the update appears to rely on oracle feedback from the stronger verifier, I wonder whether this procedure may be costly compared to simply using the strong verifier at test time.
* Is this method sensitive to the choice of the weak verifier and generator?

**Limitations:**

No, the author does not include the limitation section.

The empirical evaluation could be further strengthened. The paper would benefit from a more detailed discussion of how the algorithm can be deployed in practical settings, including how thresholds transfer to new data and what computational overhead is incurred during online updates. Finally, comparisons with a wider set of relevant baselines would make it easier to assess the method’s advantages over existing alternatives.

**Strengths And Weaknesses:**

Strength
1. The paper provides a detailed theoretical analysis and insights.
2. The paper includes empirical evidence supporting the proposed approach.
3. The paper is well organized and easy to follow.

Weakness
1. The overall contribution appears somewhat incremental. The core idea of adaptively setting thresholds for weak and strong verifiers seems closely related to existing work, which limits the perceived novelty of the paper.
2. The paper compares primarily against two extreme baselines, while leaving out other calibration-based baselines. This makes it harder to assess whether the proposed method offers substantial advantages over simpler alternatives.

---

> ### Author Rebuttal · Authors · 2026-03-27
>
> We thank the reviewer for the thoughtful comments and helpful suggestions. We are glad the reviewer found the paper theoretically detailed, empirically supported, and clearly presented. We address the questions and comments made by the reviewer.
> ## Q1.
> Our method is specifically designed to handle changing data streams. The online analysis is fully distribution-free and does not rely on any distributional assumption on the sequence of problems, the generator, or the weak verifier (Theorem 5.1). Accordingly, the guarantee is not about learning a fixed threshold on one dataset and hoping it transfers unchanged to a new domain; rather, the thresholds are updated online as the stream evolves. If a major distribution shift occurs, there may naturally be a transient adaptation phase while the thresholds adjust to the new regime. However, this is exactly the setting covered by our theory: the online guarantees continue to hold for arbitrary streams, and the adaptation behavior is controlled by the finite-sample bounds. We will clarify this point explicitly in the revision.
> ## Q2.
> The updates do rely on feedback from the strong verifier, but this is inherent to the problem setting rather than a drawback of the method. Our framework is motivated by situations where strong verification is the most reliable option available, such as expert human verification or a much more expensive inference-time procedure, while weak verification is much cheaper but noisier. If one can afford strong verification at every step, then of course that is the ideal solution. The point of our method is precisely to avoid paying that cost on every instance: SSV invokes the strong verifier only when needed, and otherwise relies on the cheaper weak verifier, while still controlling type-I and type-II errors. Moreover, the update itself is computationally lightweight, since it only updates two scalar thresholds when strong feedback is observed. We will make this deployment perspective clearer in the paper.
> ## Q3.
> From the theoretical standpoint, no: our guarantees do not require any assumption on the quality of the weak verifier or the generator. From the practical standpoint, however, the quality of the weak verifier naturally affects how much strong-verification cost can be saved. If the weak verifier provides little useful signal, then there is no alternative but to rely more heavily on the strong verifier. In this sense, the role of SSV is not to make a weak verifier strong, but to reduce the use of the strong verifier whenever the weak one is sufficiently informative on some instances. The quality of the generator plays a different role: it affects whether the underlying task can be solved at all, rather than the reliability-cost tradeoff of verification. If the generator is very poor, then most responses will be rejected regardless of whether weak or strong verification is used. We will add this distinction to the revised discussion.
>
> **To directly address this concern, we conducted two additional experiments comparing generators of varying capability and weak verifiers of varying quality. A summary of results and findings: https://imgur.com/a/cpJfity (generators) and https://imgur.com/a/B4dXxSl (verifiers). Additional results for all datasets will be included in the revision.**
>
>
> ## On weaknesses.
> The paper contributes: (i) a formal weak-strong verification framework for reasoning systems, (ii) a population-level analysis identifying the optimal two-threshold structure and the role of calibration and sharpness, and (iii) a finite-sample online algorithm with distribution-free control of type-I and type-II errors under partial feedback. We will revise the paper to make this distinction more explicit.
>
> For the rest, we note that the paper already includes a comprehensive related work section where we discuss connections to prior work and clearly explain how our setting and guarantees differ. To the best of our knowledge, we are not aware of an existing baseline that can be directly applied to our setup, i.e., one that provides distribution-free online adaptation of weak and strong verification, treats all modules as black boxes, and aims to control type-I and type-II errors with provable guarantees. If the reviewer is aware of such a baseline, we would be happy to include and compare against it.
>
> That being said, the two extreme baselines are carefully designed to showcase the two ends of the reliability vs. cost tradeoff, and thus allow us to clearly observe how effectively SSV interpolates between them.
>
> ## On limitations.
>
> We would also like to clarify that the paper does include a limitations discussion (Section 7). In particular, we explicitly note that the current policy depends only on the weak score and therefore provides marginal rather than contextual control. Due to space constraints, this discussion was kept brief, but we will expand it and make it more prominent in the revised version.

---

> > ### Author Rebuttal · Reviewer_xjLV · 2026-04-03
> >
> > The authors address my concern. I would like to raise my score to 4.

---

### Official Review · Reviewer_cSij · 2026-03-13

**Soundness:** 3
**Presentation:** 4
**Significance:** 3
**Originality:** 3
**Overall Recommendation:** 4
**Confidence:** 3

**Summary:**

This paper studies the problem of orchestrating weak and strong verification signals in LLM reasoning systems. It formalizes weak–strong verification policies, introduce metrics capturing type-I errors (incorrect acceptance), type-II errors (incorrect rejection), and strong-verification frequency. It also proposes a new algorithm called Selective Strong Verification (SSV) and provides distribution-free, finite-sample guarantees on empirical error rates when given arbitrary query streams. Experiments on mathematical reasoning and step-by-step Sudoku shows that SSV can approach strong-only reliability with fewer calls to the expensive strong verifier.

**Compliance With Llm Reviewing Policy:**

Affirmed.

**Final Justification:**

I keep my positive score 4, and encourage the authors to consider more benchmarks and experiments comparing to existing L2D approaches for a broader practical evaluation.

**Key Questions For Authors:**

- please provide any further info/context on the weaknesses mentioned above
- Figure 3 not mentioned anywhere in the text (probably should be in 6.2 ? where I wasnt sure which results are being discussed)
- Did you test different streams and how task distributions may affect things?

**Limitations:**

yes

**Strengths And Weaknesses:**

Strengths
- The paper addresses an important and interesting problem: how to reliably combine cheap but noisy verification with expensive but robust verification in LLM systems. The weak–strong abstraction is clearly defined, and the introduced metrics (type-I, type-II errors, and SV frequency) and notions of  calibration and sharpness provide a  a good formalization to reason about reliability–cost tradeoffs.

- The SSV algorithm is presented well and supported by good theoretical guarantees. The distribution-independent and finite-sample guarantees under arbitrary query streams distinguish this work from standard selective prediction or learning-to-defer approaches that typically assume i.i.d. data.

- The paper is well organized and easy to read, and provides clean formalization, theoretical results and empirical evaluation..




Weaknesses

- The evaluation is restricted to two tasks (MATH and 4×4 Sudoku) and a single generator–weak verifier–strong verifier stack. There is no exploration of robustness across different LLM families, weak verifier qualities, or stream orderings. Thus the empirical evidence mainly serves as proof-of-concept rather than demonstrating broad practical generality.

- The most significant reductions in strong-verifier calls are observed in easier settings, particularly Sudoku and lower-difficulty MATH problems. For harder MATH problems, improvements in SV reduction are relatively limited, possibly indicating that the benefits of SSV depend strongly on weak verifier sharpness. This raises questions about practical efficiency in challenging reasoning domains.

- The paper does not compare SSV against learning-to-defer or selective prediction baselines, making it unclear whether the proposed method offers practical advantages beyond its theoretical guarantees. Without such comparisons, it is difficult to assess whether SSV achieves better accuracy–cost tradeoffs than existing deferral approaches.

---

> ### Author Rebuttal · Authors · 2026-03-27
>
> We thank the reviewer for the positive assessment and thoughtful feedback. We are glad the reviewer found the problem important, the weak–strong abstraction and formalization clear, the theoretical guarantees strong, and the overall presentation well organized and easy to follow. We also appreciate the constructive comments on the empirical evaluation and will use them to strengthen the paper. Below we address the raised points and indicate the revisions we will make.
>
> **On empirical breadth and robustness**
>
> We agree that the current experiments are best viewed as a proof of concept, and that broader empirical coverage would strengthen the practical picture. **In the revision, we are adding experiments with a wider variety of weak verifiers, strong verifiers, and generators.**
>
> We compare multiple generators of varying capabilities on our benchmarks and show how generator quality impacts the Pareto curve and how SSV takes advantage of better generator quality to achieve better accuracy–cost tradeoffs. **A summary of results across all MATH difficulty levels is available here: https://imgur.com/a/cpJfity**
>
> We would also like to note that although the current evaluation uses two datasets, the empirical scope is somewhat broader than that description may suggest, since the MATH benchmark itself contains multiple reasoning task types and difficulty levels. Still, we agree that expanding to additional datasets would be valuable. For the camera-ready version, we plan to broaden the dataset coverage further as well. At the rebuttal stage, however, setting up and validating a new reasoning dataset pipeline from scratch is somewhat time-consuming, so our immediate priority is to strengthen the current empirical section through additional model/verifier ablations.
>
> **On the reduced gains in harder MATH problems.**
>
> We agree with the reviewer that the reduction in strong-verifier usage is smaller on harder MATH problems. More broadly, the quality of the weak verifier is indeed crucial, and there is no free lunch here: if the weak verifier does not provide sufficiently informative signals, for instance because it has poor sharpness, then heavier reliance on the strong verifier is unavoidable. Importantly, in this work we are not directly trying to design the best possible weak verifier. Rather, our goal is to understand what properties of a weak verifier make it useful, and to develop an algorithm that can make the best possible use of a given weak–strong pair. In this sense, the paper is complementary to the line of work that focuses on designing better cheap checks or weak verifiers. We will make this distinction clearer in the revision and highlight it more explicitly as a future-work direction.
>
> Furthermore, we have conducted experiments comparing additional weak verifier signals and show that lower weak verifier quality does indeed translate to worse achievable tradeoffs. One can therefore expect SSV to achieve better tradeoffs as weak verifier signals improve. A summary of these results is available here: https://imgur.com/a/B4dXxSl
>
> **On comparisons to learning-to-defer or selective prediction baselines.**
>
> We appreciate this suggestion. The paper already discusses the relation to selective prediction and learning-to-defer in the related work section, but we agree that an empirical comparison would be helpful if a directly applicable baseline is available.
>
> Existing l2D and and selective prediction methods are related to our setting, as we discuss in the related work, but are not directly comparable empirically. The key differences are that our framework operates online with distribution-free simultaneous control over both type-I and type-II errors under partial feedback, whereas existing baselines typically require offline training data to learn deferral policies and control a single error criterion. We will clarify this distinction further in the revision.
>
> **On Figure 3.**
>
> Thank you for catching this. We apologize for this oversight and will fix it in the revision.
>
> **On different streams and task distributions.**
>
> This is a very good suggestion. Since our theory is explicitly designed for arbitrary sequential streams, it is useful to better illustrate this point empirically as well.
>
> To address this , **we have conducted additional experiments across two types of distribution shifts in the online stream.**
> 1. First, we vary the underlying task examples themselves and consider all possible orderings of the three MATH difficulty levels and show that our method converges to the target error rates under all stream orderings. Results are available here: https://imgur.com/a/hssp5rx.
>
> 2. Second, we introduce a weak verifier distribution shift mid-stream and show that our algorithm maintains and converges to the target error rates, with the threshold evolution under the shift also visualized. Results are available here: https://imgur.com/a/OK5SOUY. We will include all of these additional results in the revision.

---

> > ### Author Rebuttal · Reviewer_cSij · 2026-04-04
> >
> > Thank you very much to the authors for the detailed response and the additional experiments that provide further demonstration of empirical gains. I encourage the authors to consider more benchmarks and experiments comparing to existing L2D approaches for broader empirical evaluation (while there may be theoretical distinctions from existing L2D methods, I think it would still be informative to include a comparison under some controlled settings with the same weak/strong stack to contextualize the practical efficiency of SSV). I keep my positive score.

---

### Decision · Program_Chairs · 2026-04-30

**Decision:**

Accept (spotlight)

**Comment:**

This paper formalizes weak-strong verification as a distinct layer for LLM reasoning systems and introduces SSV, an online method with distribution free control of incorrect acceptance and incorrect rejection relative to the strong verifier. All reviewers gave a positive score, with 4455. The rebuttal has addressed many of the reviewers' concerns and all reviewers chose the option "fully resolved". I support accepting this paper.